# Review of EEG Affective Recognition with a Neuroscience Perspective

**DOI:** 10.3390/brainsci14040364

**Published:** 2024-04-08

**Authors:** Rosary Yuting Lim, Wai-Cheong Lincoln Lew, Kai Keng Ang

**Affiliations:** 1Institute for Infocomm Research, Agency for Science, Technology and Research, A*STAR, 1 Fusionopolis Way, #21-01 Connexis, Singapore 138632, Singapore; lim_yuting_rosary@i2r.a-star.edu.sg (R.Y.L.); leww0001@e.ntu.edu.sg (W.-C.L.L.); 2School of Computer Science and Engineering, Nanyang Technological University, 50 Nanyang Ave., 32 Block N4 02a, Singapore 639798, Singapore

**Keywords:** EEG emotion recognition, neuroscience, event-related potential, evoked oscillations, valence, arousal, affective computing, emotional intelligence

## Abstract

Emotions are a series of subconscious, fleeting, and sometimes elusive manifestations of the human innate system. They play crucial roles in everyday life—influencing the way we evaluate ourselves, our surroundings, and how we interact with our world. To date, there has been an abundance of research on the domains of neuroscience and affective computing, with experimental evidence and neural network models, respectively, to elucidate the neural circuitry involved in and neural correlates for emotion recognition. Recent advances in affective computing neural network models often relate closely to evidence and perspectives gathered from neuroscience to explain the models. Specifically, there has been growing interest in the area of EEG-based emotion recognition to adopt models based on the neural underpinnings of the processing, generation, and subsequent collection of EEG data. In this respect, our review focuses on providing neuroscientific evidence and perspectives to discuss how emotions potentially come forth as the product of neural activities occurring at the level of subcortical structures within the brain’s emotional circuitry and the association with current affective computing models in recognizing emotions. Furthermore, we discuss whether such biologically inspired modeling is the solution to advance the field in EEG-based emotion recognition and beyond.

## 1. Introduction

Humans are social beings highly dependent on partnerships and alliances of varying sizes and scales to survive in a world of scarcity. At the heart of every successful formation of social partnerships is the ability to effectively recognize and identify socially relevant emotions of the self and that of others to guide and direct our social interactions. Emotions can be defined as the innate biological state of being that, for the most part, is associated with an individual’s feelings and cognitive states—an internal evaluation and reaction in response to perceived events stemming from the interaction with various stimuli from the external world [1,2,3]. From this, observable behavioral changes influenced by the experienced emotional states include changes in facial expressions, vocal intonations, and bodily gestures [4,5].

For several decades, studies and findings pertaining to emotions have garnered the interest of many. The most recent WHO statistics reported that in 2019, 970 million people globally suffered from an emotion-related mental disorder, especially for anxiety and depression, which exhibited a 26% and 28% increase within a year, respectively [6]. Acknowledging the importance of studying and detecting emotions, a pioneering group of researchers began the field of work by categorizing emotions into six fundamental classes: happiness, sadness, anger, disgust, surprise, and fear [7,8,9], which can be distributed along the scales of valence and arousal values [10,11,12,13] as depicted in Figure 1 (the six fundamental classes are denoted in colored font). Apart from the Circumplex model, other models such as the Pleasure–Arousal–Dominance (PAD) model and the Five Factor Model (FFM) exist as well for emotion classification [14]. However, the use of other models, such as the PAD model, with the inclusion of a third dimension, “Dominance”, has been deemed to be more suitable for studying nonverbal cues and communication as commonly seen in body language in the field of psychology instead [15]—which lies outside the scope of our review article. The use of the Circumplex model in this article is neither (1) proclaiming that it is the model superior to other existing models for the description of emotions nor (2) demonstrating that it is the only model able to provide an exhaustive description of all known emotions. The sole purpose of the Circumplex model is merely as a simple classification of the six fundamental emotion classes that is aligned with the purpose and scope of our review.

Some of these emotions have been shown in neuroscience literature to be generated by specific brain structures [16] and are then either consciously detected and recognized by the individual from whence they arise or detected by an external device via physiological and/or physical signals. With technological advancement, the detection of emotions and the collection of their associated data are now less subjective, as it is no longer necessary to depend solely on qualitative measures such as self-reports and questionnaires. Some widely used objective modalities for detecting the generation of emotions include the electroencephalogram (EEG), electrodermal activity (EDA), electrocardiogram (ECG), and electro-oculogram (EoG) [17,18,19].

In particular, the utilization of electroencephalogram (EEG) signal readouts for the detection and recognition of emotions is present in a plethora of emotion and affective computing literature. While other modalities of detection and measurement have their own drawbacks as a standalone [17], the EEG modality is one of the most commonly used to detect and recognize the generation of emotions because of its high degree of objectivity in obtaining intrinsic, involuntary emotional responses as opposed to physical readouts such as those from facial expressions or bodily gestures [20]. Therefore, some recent studies have used a combination of EEG with other modalities of detection e.g., EDA to bolster findings on how emotions can be better recognized and characterized with correlations found across those modalities [21,22].

Nonetheless, studies have posited that emotional states are correlated to the brain’s electricity activity and therefore can be quantitatively detected and measured using EEG devices. By virtue of EEG signals with characteristics of excellent temporal resolution and the capacity to detect electricity activity from various brain regions with varying numbers of EEG electrode channels, this modality allows for the real-time detection of emotions constituting an objective and reliable assessment of an individual’s internal state. The crux would then be how one would go about selecting which spatial region(s) of the brain (i.e., the specific electrode channels) to focus on when analyzing the EEG data collected for emotion recognition, the specific features (i.e., EEG signal signatures, a specific or a range of frequency bands) to extract at certain epochs along the temporal dimension, and whether those features are robust over time.

Of note, some recent works have proposed using a spatiotemporal model to capture both the information of different channels (spatially) and at different time stamps (temporally) [23], which will help to ensure that the association of the other brain regions and at different temporal time EEG signals will be learned by the model. Furthermore, in addition to different modeling, the research on EEG-based emotion recognition has undertaken different feature extraction processes on the EEG signals [24,25,26]. In contrast, others have considered specific channels [27] and investigated the frequency bands that contribute more to this recognition process. Furthermore, in considering the significant EEG signal variance among different subjects, a domain adversarial network has been proposed to incorporate into the modeling to improve the recognition by teaching the model how to identify subject-invariant features [28].

While research in the neuroscience field has provided some evidence on specific brain regions, processes, and mechanisms that are implicated in the generation of different classes of emotions, it is impossible to pinpoint one specific mechanism of a particular brain region(s) to explain for emotion recognition. On the other hand, research in the affective computing field is constantly spearheading advances in emotion detection and recognition but has no single algorithm or a computational network that can explain where and how emotions come about. Hence, the combined effort and inspiration obtained from both research spheres are vital for a holistic understanding of emotional states before the next breakthrough in this field.

Given that emotions have varying degrees of valence and arousal values, there will be times when certain emotions get so intense that they overwhelm the individual. Emotion regulation defines one’s ability to manage such overwhelming emotional episodes using strategies such as down-regulating negative emotions and/or up-regulating positive emotions [29]. Failure in emotion regulation and management is implicated in psychiatric disorders such as major depressive disorder, excessive anger and aggression, and some behavioral symptoms observed in schizophrenia. To date, there is no established algorithm or series of computational networks that is involved in the research area of emotion regulation, which is potentially the next step to elucidating and understanding a key aspect of emotional intelligence.

To better aid the reader’s expectations for this review article, Figure 2 depicts the flow of the article in sequential order.

Henceforth, this review aims to discuss the following, with evidence and perspectives from both the neuroscience and affective computing fields: (1) the spatial, temporal, and spatiotemporal aspects of emotion recognition, (2) a brief overview of the mechanism by which the brain potentially processes emotions and some recent coverage of related work from the neuroscience field, and (3) the current standing for state-of-the-art computational models for emotion recognition and whether bio-inspired modeling is the key to the next breakthrough in affective computing such as in the domain of emotion regulation.

## 2. Methodology of the Review

The inclusion and evaluation of papers in this review follow the preferred reporting items for systematic reviews and meta-analysis (PRISMA) guidelines. Research on neuroscience was sourced using the following keywords and the PubMed search engine: <emotion recognition, or regulation, or processing neuroscience>; <emotional regulation computational>; <visual emotion recognition>; <emotion recognition or regulation EEG>; <brain regions for emotions>; <brain event related potential or erp emotion recognition>; and <brain evoked oscillation emotion recognition>. The following keywords were used to search Google Scholar for EEG emotion-based recognition research: <eeg emotion recognition, or seed eeg, or deap eeg, or gnn eeg, or spatiotemporal eeg emotion recognition>. The literature survey was initiated on 25 April 2023. As search engines such as ScienceDirect and Google Scholar Literature yielded an immense number of returned results (numbers on the order of hundreds of thousands), only a certain number of highly cited papers were selected to be part of this review. The excluded articles included patient studies, reviews/meta-analyses/methods, duplicate articles, non-keyword-matched articles, irrelevant articles, and socially related emotions e.g., pain, empathy, authenticity, etc. A total of 12 articles met the inclusion criteria for brain regions of affect, 8 articles for event-related potential, and 7 articles for evoked oscillations.

The initial search returned 200,000 papers from Google Scholar. In accordance to PRISMA protocols, we performed the following selection and review steps illustrated in Figure 3 for EEG emotion recognition. The top 100 relevant records ranked in the search engine across the searched keywords were considered for eligibility in the first round of screening. This screening filtered out non-peer-reviewed publications and those that did not fulfill the examination of the experimental protocols, such as publicly available datasets and non-subject bias evaluation methodologies. Lastly, 41 more recent modeling methodologies from 2016 and after that underscored the novelty of the approach were considered for reporting in the review.

## 3. Components of Emotions

### 3.1. Spatial Specificity of Emotions

An emotion can be broadly defined as a subconscious, transient alteration in one’s internal state evoked by a transient external event that is affectively impactful. It constitutes an orchestrated response from multiple systems including physiological and physical reactions. In neuroscience, the “spatial components” of emotion can be defined as the brain’s neural structures, nuclei, or sets of neuronal cell populations that, along with the associated processes and signaling pathways, give rise to the modulatory effects in response to emotional stimuli. Spanning previous centuries to the present, the exact biological basis or substrates of emotion in the human brain remain contentious issues in the fields of neuroscience and neuropsychology. Multiple studies have attempted to locate the exact spatial coordinates of neural bases of emotion but could not provide consistent evidence to demonstrate that specific classes of emotions are generated by the same distinct group of subcortical nuclei or regions in the brain. Examples of such studies are tabulated in Table 1.

As seen from Table 1, multiple overlapping brain regions have been shown to be activated for different classes of emotions, e.g., the amygdala. This is not exactly surprising since it would be energetically expensive to have dedicated brain bodies wired specifically for the sole function of one class of emotion. The more interesting question is as follows: how do these brain bodies, or groups of neurons within these bodies, manifest differentiated roles for the generation of neuronal activity that is associated with the respective emotion classes?

For instance, the amygdala, conventionally found to be activated in the presence of fear-associated stimuli, has recently been proven to function as valence-encoding nuclei via various chemogenetic and optogenetic experiments in animal model studies [42,43,44,45]. One study conducted in a mouse model demonstrated, via specific gene markers, that the neuronal populations within the amygdala, specifically the basolateral amygdala (BLA), contains both positive- and negative-valence-encoding neurons that are both genetically and spatially distinct [43]. R-sponin-2-expressing (Rspo2+) neurons, localized in the anterior BLA, encode negative-valence stimuli and modulate negative-valence-associated behaviors and memories. In contrast, protein phosphatase-1 regulatory inhibitor subunit 1B-expressing (Ppp1r1b+) neurons, localized in the posterior end of the BLA, encode positive-valence stimuli and modulate positive-valence associated behaviors and memories [43]. The authors also found that these two subpopulations of neurons in the BLA are mutually inhibitory populations and that they achieve this via feed-forward inhibition mediated by local inhibitory interneurons. These are keynote findings that show, in ways that fMRI cannot in terms of neural population specificity, the gating mechanism of the heterogeneous population in the amygdala that is postulated to facilitate processing of both positive and negative valence in various classes of emotions. Hence, this provides strong support to justify the amygdala being implicated in processes of more than one class of emotion as reported by several others.

The prefrontal cortex (PFC), as evidenced in Table 1, to be involved in the processing of multiple emotions, is also another crucial brain region in the emotional network. Studies have postulated that, with co-activation from the amygdala, it is involved in learning both the emotional and motivational aspects of a given stimulus so as to promote downstream signalers for top-down-driven behaviors such as feeding satiety [46]. The PFC has been shown to exhibit roles that perform very much like an airport’s control tower in terms of cognitive control. It is tasked with maintaining goal-specific patterns of activity and fine-tunes decision-making processes in realizing them. The neuronal population of the PFC provides gating signals throughout the majority of the brain, modulating multisensory modalities and networks responsible for motor execution, memory retrieval, emotion recognition, and so on. Ultimately, the net effect of these gating signals is to direct neural activity and, hence, the transmission of dominant information along network pathways that facilitate the change required for specific inputs to manifest into correct outputs to realize a certain goal. Such modulation and control from the PFC is essential—especially in cases of ambiguous stimuli where several input representations exist or when more than one competing output response is possible. Given the possibility of having conflicting inputs and competing outputs, the myriad of PFC modulatory mechanisms—whose purpose is to resolve conflict and competition and activate the appropriate neural pathways for information transmission needed to perform the task—function as the neural implementation of attentional biases, rules, or goals that are subjected to the target of their modulatory consequence. Since there exist multiple conflicting representations of inputs to the PFC, as well as multiple competing outputs as a result, the PFC must function in a way whereby its activity remains robust against constant distractions until the targeted goal is achieved [47,48,49] while maintaining its versatility in updating current representations when necessary. It must also store the appropriate representations, those that can select the neural pathways needed for the task for both long and short terms [47,50,51,52]. Hence, not only must the PFC be able to access and regulate extensive information in various brain regions but also its representations must possess the potentiality for multimodality and integration as well. In order to achieve that, the PFC itself must be highly plastic to integrate new goals and learn strategies for realizing those goals [53,54]. Apart from its extensive association with brain regions providing sensory inputs such as occipital, temporal, and parietal cortices [55,56,57], the PFC also has anatomical connections with brain regions that are sites of input integration. Regions like the rostral superior temporal sulcus and the arcuate sulcus region are sources of input to the PFC that convey information containing visual, auditory, somatosensory, or a combination of two or more components [58,59]. For disseminating and orchestrating executive motor commands for meaningful behavior, the PFC—dorsal PFC to be specific—has connections with motor areas of the brain, namely, the supplementary motor area, pre-supplementary motor area, rostral cingulate, cerebellum, and superior colliculus [60,61], which then relay information to the primary motor cortex and spinal cord. At the same time, direct and indirect connections of the orbital and medial PFC with the medial temporal limbic nuclei bodies—hippocampus, amygdala, and hypothalamus—serve the function of long-term memory storage and retrieval as well as the processing of emotion [62,63]. Since the PFC is a major hub involved in the processing and transmission of emotion information and the processing of emotion involves other brain regions subserving different cognitive functions, ‘emotion’ and the process of emotion recognition inevitably becomes inseparable from the effects of top-down and bottom-up processes/outcomes stemming from indirect connections from the attentional and memory-related networks via the PFC (and other sites of signal integration as well).

As the scope of this review is not meant to cover the extensive anatomical and functional significance of each brain region implicated in emotion recognition, only a couple of examples such as the amygdala and PFC are raised to show (1) that it is a fallacy to believe in the notion of having dedicated brain regions, nuclei bodies, or even specific neural populations for a specific class of emotion and (2) that emotions, by virtue of being processed by several integrative sites for multimodal and multisensory inputs, sit in a closely woven brain network that ties in other cognitive functions such as executive (motor) control, memory, and attention. Controversy that surfaced from neuroimaging literature wherein some studies supported the notion of discrete neural structures for emotions [16,64] while others found little to no evidence to conclude the same [65,66,67] is evidence that dedicated brain regions for specific classes of emotion are unlikely to exist. Although several studies have identified unique brain regions and/or structures for certain classes of emotion [16,64], these brain structures are not entirely self-sufficient, i.e., they are still dependent on brain regions/structures whose activation have been observed for other classes of emotions as well. Hence, it can be concluded that there is no unique group of brain regions/structures that exist solely for the purpose of driving activity for a particular class of emotion.

At this juncture, we ask the following: is there a unique mode of communication within this vast network wherein we can distinguish one class of emotion from another via a temporal sequence of events in specific brain structures? Given the highly interconnectedness of the brain’s emotional network, as seen in the case of the PFC, the next question to be raised is as follows: how does each node in the network, be it one as large as the orbitofrontal cortex of the PFC or one as small as a group of several tens of neurons in the amygdala, transmit information within the network such that only specific structures or nuclei are activated to signal a change in internal states (i.e., emotions)? The answer lies in the brain’s oscillations, which are covered in the next subsection.

### 3.2. Spatiotemporal Specificity of Emotions

In neuroscience, the “temporal specificity” of emotion can be defined as a neural signature (in the form of electrical signals) occurring over a specific time window from one brain structure to another in a sequential manner that is characteristic of a particular class of emotion, e.g., happiness [68]. These emotion neural signatures must be unique and reproducible in order to be reliably identified and distinguished from that of other classes of emotion. Since neural signatures are only termed as meaningful signals when they are transmitted from one specific region to another, it is difficult (and somewhat illogical) to separately discuss spatial and temporal aspects of communication within the brain [69]. Hence, this section elaborates on and discusses the spatiotemporal aspects of the brain as a whole before attempting to make sense of what we can interpret in the domain of emotions. As the term suggests, “spatiotemporal specificity” of emotions is made up of two components. Spatial components refer to the brain structures whose activation facilitates the modulation of the emotional experience, and temporal components refer to the mechanism related to how several brain structures activate or co-activate progressively over time to modulate a particular emotional experience [69]. In order words, it refers to the temporal code of activity for which different brain structures or populations of neurons relay information from source to receiver. This process of information transmission can be recorded to demonstrate how several brain structures work together in a sequential or parallel manner within a particular functional network [70].

Quoted from the famous architect Louis Sullivan, “function follows form” [71]. To understand the function of something, one must first decompose its form and then consider its basic units. Hence, in order to understand how the brain selects important information to be communicated downstream for emotion processing, we need to briefly understand the constituents of the brain and the mechanism of their operations.

#### 3.2.1. How Are Brain Signals Generated?

The human brain houses approximately 100 billion neurons with about 200 trillion connections among them. For brain-guided decisions that are complex enough to direct any organism, much less a human, all of those neurons and their connections need to be compacted and packed to fit into tissue matter the size of approximately 1300 cubic centimeters [72]. On top of which, these neurons need to be organized in a way where meaningful functional groups are appropriately arranged in terms of their physical sites as well as the synaptic distances between them and other neuronal populations with which they share common functions. The majority of such arrangements are a combination of short- and long-range connections that connect one neuron to another from one end of the brain to the other in an energetically optimal way [73,74,75].

However, the pure reliance on physical connections, whether directly or indirectly connecting one neuron to another, is still not computationally optimal. This is because (1) neurons are not perfect, and errors do ensue along their axons when they propagate; (2) there exist conduction delays just like any wiring system we use in our day-to-day activities; (3) the brain is not a mere feed-forward hierarchical system where information is simply pushed forth via bottom-up mechanisms. To see the brain as a simply bottom-up structure assumes that it has a terminal “almighty” layer/region that ultimately integrates all incoming information and decisions to pass a final judgment and guide outward behavior. In reality, that information does loop back to those upstream regions to exert top-down-driven influences so as to modulate or fine-tune those transmitted signals [47,48,49]; and (4) the brain, in the absence of inhibition, goes into a series of excitation outbursts with any given stimulus as in the case of epileptic scenarios. Without a control mechanism, such excitation is transmitted in a widespread manner throughout the whole brain, yet unable to provide meaningful information to decode for. Therefore, on top of an efficient network of physical connections, an overlaying network of inhibitory neurons is necessary to function as the “control system” to restrict the degree of excitatory transmissions [76,77,78,79,80].

Not only do inhibitory neurons help to control the excitatory spread in the brain but also their activity facilitates a certain rhythm of synchronous excitatory spikes that, depending on the size of the network that is recruited in a temporal window to fire in harmony, can create transient or prolonged periods of orchestrated neuronal activity observed in EEG readouts as “brain oscillations”. Such oscillatory effects vary in frequencies and amplitudes depending on the characteristics of the excitatory and inhibitory neurons recruited at a certain time point (dependent on the stimulus type and stimulation duration) [81]. By virtue of the fact that inhibitory neurons control the extent of excitation throughout the brain, they help to determine both the spatial and temporal limits for the transmission of information. In other words, such a control mechanism is able to provide functional segregation/grouping of specific brain structures with activities that are likely to be coordinated in a given temporal window when a particular stimulus is presented.

As this review is not meant to delve deep into the molecular mechanisms employed by different subtypes of inhibitory neurons to differentially control and modulate for the oscillation frequencies commonly seen in EEGs (e.g., delta, theta, and alpha waves), the interested reader is encouraged to review the following work for some basic introduction on inhibitory neurons and their neuromodulatory molecules, i.e., GABA and GABA receptor expression profiles: [82,83,84,85,86,87,88]. Briefly, the frequency of the generated oscillation is founded on the mean duration of inhibition exerted by the inhibitory neurons. For instance, if the inhibition is facilitated by GABAA receptors, which are fast-acting, the consequential excitatory activity that it modulates for will be reflected as a gamma frequency (40–100 Hz) [89,90,91].

The coordinated activities detected in specific brain regions via a scalp EEG are reflected as brain oscillations, and this is how the brain—via finely-timed rhythmic activity of excitatory neurons over various time scales—coordinates various regions/structures to communicate swiftly and effectively. From this, it can be seen that the exact or “correctness” of how neurons are wired together in the brain falls secondary in importance to how different groups of neurons sync their activities and action potentials in response to the generated rhythm via the inhibitory network.

#### 3.2.2. What Information Is Transmitted in Brain Signals?

Before we attempt to decipher the information that is transmitted in oscillating brain signals, we need to understand how these brain oscillations engage their targeted neural populations.

In terms of temporal precision, it is common to require hundreds of neurons to fire cooperatively together where the coordinated activity reaches their targets at specific arrival times in order for the transmission to be meaningful [92,93]. In terms of spatial precision, information is often sent out in a parallel manner over different brain regions for simultaneous processing [94,95]. Hence, it has been postulated that different frequency bands function to recruit varying pools of neural populations for different extents of information transmission and computation in a parallel manner. Several studies have shown evidence for this hypothesis. Von Stein’s team found, using EEG, an increase in coherent activity in the 13–18 Hz beta frequency range between the temporal and parietal region across three different modes of presentation, i.e., visual presentations via pictures and written words separately and auditory presentations via verbalization. This increase in coherent beta activity is only observed during trials where semantically meaningful stimuli in all three presentation modes were shown as opposed to when nonsensical words, verbalizations, or line drawings were presented [96].

Another EEG study investigating the changes in signal coherence during a working memory task found a significant enhancement in the 4–7 Hz theta frequency between the prefrontal and posterior brain regions during 4 s retention intervals. They also found increased synchronization in the 19–32 Hz gamma band during the perception and retention epochs. However, they rationalized that gamma activity—a hallmark of synchronization in local neural circuitry—is primarily involved in sensory processes, while inter-regional brain interactions implicated in tasks such as working memory would recruit neural populations on a larger scale [97]. They had suggested that the presence of gamma band activity during a working memory task helmed by theta activity may imply a nested function of gamma within theta: an idea of cross-frequency coupling that was later investigated by another group of authors whose experiments were designed to uncover a potential function of having fast oscillations nested into slow oscillations.

Alekseichuk and his team executed a tACS (transcranial alternating current stimulation) delivery of high-frequency gamma bursts (80–100 Hz) superposed onto theta oscillation to the left parietal region during the encoding phase and showed that it facilitates the enhancement of spatial working memory performance. This phenomenon was only observed when the high-frequency gamma bursts were superposed onto the peaks, but not the troughs, of the theta oscillation [98].

In the realm of animal studies, Fujisawa and Buzsáki have found, using an M-maze turn choice working memory task in rats, that a 4 Hz theta oscillation strongly influenced neural activity in the PFC–VTA (prefrontal cortex-ventral tegmental area) circuitry. In addition, this PFC–VTA theta oscillation was phase-coupled to the theta oscillations of the hippocampus when working memory was active. Local gamma oscillations within the PFC–VTA circuit were also phase-locked to the established 4 Hz oscillation during working memory [99].

Taken together, these studies demonstrate that slower oscillations such as theta are capable of recruiting and engaging a larger network of different functional populations of neurons that are spatially further apart, while faster oscillations such as gamma are present in localized brain regions or networks to modulate transient processes. Given that such findings exist in animal models such as rodents, this phenomenon is shown to be evolutionarily conserved.

In the domain of emotions and emotion processing, studies have also been conducted to investigate the correlations between dynamic changes in brain oscillations detected during emotion processing or when emotion-evoking stimuli/content are presented. In a nutshell, the neurophysiological signals that can be detected from the brain to be analyzed for emotion recognition, or any emotion-related work, can be broadly categorized into two groups: event-related potentials (ERPs) and brain oscillation frequencies, which some literature has coined evoked oscillations (EOs).

##### Insights from ERPs for Emotion Recognition

With regards to studies using ERPs as a biomarker to investigate emotions, some studies have utilized the onset of ERPs such as N170, P300, and the late-positive potential (LPP) to either draw correlations with time points of emotion recognition or to investigate the changes in the characteristics of those signals to determine the nature or properties of the emotional stimulus shown. What information can ERPs provide pertaining to emotion recognition, and are they truly reliable? Table 2 below shows a brief summary of some keynote findings that suggest the implication of ERPs in emotion recognition.

As a summary, where works that have leveraged ERPs to either detect different types of emotions [100,101], emotion valence [102,103,104,105], or changes in emotion [100] are concerned, the basic conclusions that can possibly be drawn are as follows: (1) emotional faces induce an increased positive ERP as compared to neutral faces in the early phase of perception on time scales ranging from 100 to 200 ms [106]; (2) early ERPs occurring before 200 ms have been shown to be specific to emotional face stimuli and not simply any category of emotional stimulus [107]; (3) it is unlikely that ERPs are representative characteristics of each class of emotion, namely, happiness, sadness, anger, fear, etc., neither do they correlate with some form of spatial specificity in which those emotions occur; (4) some support for the notion of hemispheric lateralization for emotional valence where the right hemisphere is deemed to be specialized in perceiving and recognizing negative-valence emotions while the left hemisphere is for positive-valence emotions; and (5) ERPs function, at best, as stimulus onset detectors or change detectors. From these, it appears that ERPs mostly function to demarcate when the brain detects a change in its attended environment—in the case of experimental settings, it is the onset of a stimulus (emotional/non-emotional).

The furthest one can argue for the localization of ERPs in terms of cortical specificity for emotions is how they signal for a change in emotion expression [100] but not individual emotion per se. Dzhelyova and team used an emotion change detection task to investigate the spatial correlates of ERPs pertaining to each class of emotion. Although broadly distributed across the occipito-temporal region, they found P100, N170, and P300 ERP signals that are more significant in the dorsal occipito-temporal region for neutral–happy change and in the ventral occipito-temporal region for neutral–fear or neutral–disgust change. All 3 types of ERPs are present across all types of emotion change. Their results also lend support to another similar observation in literature, that ERPs can be valence sensitive.

As examples, P450 and LPP have been shown to be enhanced by positive valence [102] with LPP possibly more enhanced by negative compared to positive valence [103,108]. N200 and P300 have been shown to be elicited in response to changes in emotional valence [104], and they seem to be enhanced across all modes of emotion valence change, i.e., static image changes, auditory intonation changes, and audio-visual changes [101]. N170 has been shown to elicit neither an effect for emotion valence or task but has been proven to be a face-specific biomarker [105,106].
brainsci-14-00364-t002_Table 2Table 2Studies on event related potentials (ERPs) implicated in emotion recognition.
StudyData Acquisition ModalityExtracted Biomarker(s)EEG Channels & Associated Brain Region(s)Stimulus Modality for Facial ExpressionsEmotionInvestigated Emotion Dimension(s)Major FindingProverbio  et al., 2020 [102]EEGP450, late anterior negativity response, late positivity (LPP)RIGHT and LEFT frontal regionsAudioPositive, Negative valenceValenceP450 and LP enhanced by positive valence content; anterior negativity was enhanced by negative content; negative speech activated right temporo/parietal areas; positive speech activated left homologous and inferior frontal areasBondy  et al., 2017 [103]EEGlate positive potential (LPP)Parietal regionStatic imagesPositive, Negative valenceValenceGreater LPP for emotional images relative to neutral; negative valence images elicited greater LPP than positive valence images; greater LPP in younger subjects than older subjectsDzhelyova  et al., 2016 [100]EEGP100, N170, P300RIGHT dorsal occipito-temporal region (happiness), RIGHT ventral occipito-temporal region (fear and disgust)Static imagesHappy, Fear, Disgust, NeutralType of emotion, expression change detectionUnique spatial topography for each emotion type: dorsal occipito-temporal region (happiness), and ventral occipito-temporal region (fear and disgust); time-domain EEG markers apparent across all expression changesSchirmer  et al., 2012 [104]EEGP200Frontal, central, parietal regionsVisual words, and audioSad, NeutralWord valenceIncreased P200 amplitudes to sad intonation during encoding; voice-related retrieval effects observed in P200 using test words during retrievalSchrammen  et al., 2020 [105]EEGN170, alphaTemporo-parietal region (N170), frontal region (alpha)Static imagesHappy, AngerValenceMore negative amplitudes in the dominant hemisphere symmetrical to handedness; frontal alpha asymmetry is an index of inhibitory control; right hemisphere dominance for negative valence, left hemisphere dominance for positive valenceChen  et al., 2016 [101]EEGN100, N200, P300,
alpha, beta, thetaWhole brainStatic images, audio, audio-visual pairing of static face image to audio sentenceAngry, NeutralType of emotionBimodal emotional changes detected with higher accuracy and faster reaction times; all emotional changes across modalities induced greater theta synchronization, enhanced amplitudes in N200/P300 complex; P300 and theta oscillations are important for emotional change detectionMartini  et al., 2012 [108]EEGP300, LPP, gammaLEFT/RIGHT temporal and frontal (negative valence), LEFT temporal and RIGHT parieto-temporal (neutral)Static images of scenesNegative valence, NeutralValenceIncrease in P300 and LPP components for negative valence pictures across right hemisphere; enhanced gamma for negative valence pictures; increased gamma power for negative valence pictures; early strong between-region gamma synchronizations for negative valence pictures; late strong between-region gamma synchronization for neutral picturesEimer  et al., 2003 [106]EEGN170, P200Frontocentral region (P200)Static imagesAngry, Disgusted, Fearful, Happy, Sad, Surprised, NeutralValenceN170 amplitudes displayed neither a main effect for valence or task in response to neutral vs emotional faces; sustained P200 elicited in response to emotional faces at frontocentral sites and then a broadly distributed P300 thereafter


Therefore, it can be said that none of such ERPs, whether as a single entity or a combination of those, could function as a neural signature(s) that is characteristic of emotions per se. Yet, one could also argue, using evidence from the field of invasive intracranial research, that such ERPs detected from the scalp may potentially be summed resultants of oscillating signals coming from subcortical brain structures that are specific to emotions. Multiple investigators have recorded local field potentials (LFPs) using intracranial electrophysiology during epochs of facial expression processing when subjects were presented photographs of emotion-expressing faces. With the onset of the stimulus (presentation of the photograph), initial activity was detected in the primary visual cortex around 100 ms, followed closely by activation of the amygdala [109,110], especially in the presence of fearful expressions, which occurs within the first 200 ms of stimulus onset. Subsequently, activity in the fusiform face area (FFA) ensues at around 120–200 ms [111]. In studies using dynamic emotion-expressing faces as stimuli [112], the FFA and superior temporal gyrus (STG) were found to co-activate in the temporal range of 200–500 ms from stimulus onset. Finally, the orbitofrontal cortex gets activated in the temporal window of 500–1000 ms [113].

Although these human intracranial electrophysiological studies might provide a moderate rationalization to the occurrence of ERPs—signals propagating from the respective subcortical nuclei are picked up by EEG of the scalp—it would benefit the reader to bear in mind that brain signals from those regions would have been summed, averaged, and/or subjected to interferences apart from other recording artifacts during EEG signal acquisition. Hence, it is important to recognize that ERP readouts, within reason, can be indicative of the brain’s “awareness” of an attended stimulus, but they cannot be utilized to inform where emotions are spatially localized or which subsequent downstream brain regions are recruited for emotional processing.

Given that the scalp spatial specificity of ERPs disappears after the first 200 ms from stimulus onset, more reliable signal candidates that do not dissipate over short time scales need to be further investigated on the flow of information transmission across the cortex and how that information is likely processed for correct emotion recognition. To do that, we need to consider changes in brain oscillations.

##### Insights from Evoked Oscillations for Emotion Recognition

Oscillations are a long-standing mode of communication between large groups of neural populations in the brain. The question here is the following: how do brain oscillations compel billions of neurons to cooperate in a population-specific manner across different regions?

Recall in the previous section where the network of inhibitory neurons and their function in the brain was introduced. Their main task is to act as a control mechanism to limit the extent of excitation stemming from the excitatory neurons. In doing so, they facilitate the degree and direction of information transmission throughout the brain. However, one must not forget that despite having a system of efficient connectivity among neurons, conduction delays along their axonal branches remain their major drawback in terms of computational speed [114]. Therefore, the network of inhibitory neurons will need to work cooperatively in order to compel targeted populations of excitatory neurons to do that as well. Imagine that neurons are audiences in a theater: when the performance draws to a close and the applause happens, the early phase of it would have clusters of clapping going in and out of sync with several other clusters. In the next few seconds, one would observe that those who, intentionally or unintentionally, slowed down the rhythm of their clapping, will have many others join in one after another until everyone is on board clapping to the same rhythm and pace. This phenomenon serves to show that in the presence of a slower rhythm, more individuals will be able to ‘listen in’ to the rhythm and follow along [115]. Conversely, the presence of a faster rhythm will not allow as many to join in the same rhythm as fast.

This is analogous to what is happening in the brain: oscillations of different frequencies (i.e., ‘rhythms’) exist to determine which population(s) of neurons is to be included/excluded from a frequency-banded cluster so that the appropriate group can ‘listen in’ to the appropriate information and transmit it downstream. The slower the oscillating frequency, the larger the pool of neurons that can be recruited as a large functional network. Likewise, in an opposite manner, the faster the oscillating frequency is, the smaller the pool of neurons recruited as a functional network. This is known as the inverse relationship between oscillations and the neural pool size it is able to recruit [116].

With the understanding of the inverse relationship between oscillations and the neural pool in mind, we attempt to rationalize findings found in the literature pertaining to how oscillatory signal readouts from the brain can inform on emotions. Again, we briefly discuss a few recent works as examples to each of the commonly investigated oscillatory bands that have been shown to be implicated in emotions and emotion processing, namely, the alpha, beta, theta, delta, and gamma bands. Table 3 below gives a brief summary of some findings from studies suggesting the implications of evoked oscillations in emotion recognition.




**1. Alpha Band**


Summing up the major conclusion drawn from the studies investigating the alpha band in general, the interplay of alpha with emotional processing strongly suggests an inhibitory effect on other brain regions when processing emotional stimuli, and it seems to have a valence-specific hemispheric lateralization effect.

The study conducted by Schrammen and team involved a Go-NoGo task using an angry face as the ‘Go’ target in one block of experiments and a friendly face as the ‘Go’ target in another. Both stimuli were presented in the left and right visual field in a randomly controlled manner. Their results on individual alpha power showed that subjects with enhanced right frontal alpha activity were better at inhibiting positive valence stimuli than negative valence stimuli, and vice versa for subjects with enhanced left frontal alpha activity indexed by their results from false alarm rates [105]. In a similar demonstration of the valence-specific lateralization effect on emotions facilitated by alpha frequency, Günteki and Basar showed that a higher amplitude in alpha power was elicited for angry as opposed to happy expression localized in the left posterior region. They inferred that as an inhibitory effect exerted by enhanced alpha power to allow the right hemispheric region the capacity to process the angry (i.e., negative valence) emotional expression [117].

With regards to detecting changes in emotions, Chen’s group found a significant alpha/beta desynchronization during bimodal emotional changes as compared to that in the no-change condition [101]. Along similar lines of work, Popov and team showed an increase in alpha power in bilateral sensorimotor areas preceding emotion recognition and suggested that this synchronization in alpha frequency decouples the sensorimotor face area to prevent further noise interferences for successful recognition [118].

The more substantial finding was in an invasive intracranial study conducted by Zheng and team, where they targeted recording electrodes in the amygdala and hippocampus contralateral or outside the epileptogenic region in patients. Patients were tasked to identify if the emotional images presented were ‘new’ or ‘old’ (i.e., seen in the encoding phase) in which there were similar and dissimilar foils, i.e., ‘lures’. They found that in trials where patients failed to reject the lure (‘lure-false-alarm’), there was enhanced alpha power from the amygdala, which in turn modulated a decrease in both amygdala and hippocampal theta power as well as a decrease in high-frequency hippocampal activity. On the other hand, in trials where patients correctly rejected the lure (‘lure-correct-reject’), there was decreased alpha power and therefore enhanced theta power from the amygdala. This enhanced amygdala theta power in turn synchronized with a similarly enhanced hippocampal theta power. High-frequency activity in both brain structures was then modulated by the theta phase of the other [119]. This is the first precise demonstration of the mechanism by which the amygdala acts on the hippocampus: a unidirectional influence from the amygdala alpha power modulating hippocampal theta power and high-frequency activity for successful memory retrieval of emotional stimuli. And it is yet additional convincing evidence of how enhanced alpha power serves as the inhibitory control at the interface of emotion processing and memory retrieval, which subsequently implicates decision-making and behavior.
brainsci-14-00364-t003_Table 3Table 3Sudies on evoked oscillations (EO) implicated in emotion recognition.StudyData Acquisition ModalityExtracted Biomarker(s)EEG Channels & Associated Brain Region(s)Stimulus Modality for Facial ExpressionsEmotionInvestigated Emotion Dimension(s)Major FindingSchrammen  et al., 2020 [105]EEGN170, alphaTemporo-parietal region (N170), frontal region (alpha)Static imagesHappy, AngerValenceMore negative amplitudes over the projected hemisphere according to the visual field for which the No-Go stimulus was presented; more negative amplitude in the dominant hemisphere symmetrical to handedness; frontal alpha asymmetry is an index of inhibitory control, frontal asymmetry not related to handedness; right hemisphere dominance for negative valence, left hemisphere dominance for positive valenceChen  et al., 2016 [101]EEGN100, N200, P300, alpha, beta, thetaWhole brainStatic images, audio, audio-visual pairing of static face image to audio sentenceAngry, NeutralType of emotionBimodal emotional changes detected with higher accuracy and faster reaction times than unimodal; all emotional changes across modalities induced greater theta synchronization, enhanced amplitudes in the N200/P300 complex, P300 amplitudes; P300 and theta oscillations crucial for emotional change detection; facial and bimodal change caused significant N100 enhancement and larger alpha/beta desynchronizationZheng  et al., 2019 [119]Intracranial electrodestheta, alphaAmygdala and hippocampus (contralateral or outside of epileptogenic region in patients)Static images of scenesPositive, Negative valenceValenceIncorrect response trials enhanced alpha power from amygdala modulates a decrease in both amygdala and hippocampal theta power and a decrease in high-frequency activity in the hippocampus; correct response trials decreased alpha power and enhanced theta power from amygdala synchronized with similarly enhanced theta power in hippocampus with high-frequency activity in both regions modulated by theta phase of the other; amygdala exerted a unidirectional influence on the hippocampus for memory retrieval via changes in alpha and theta band powerTang  et al., 2011 [120]EEGgamma (low and high)LEFT/RIGHT/MIDDLE frontal, LEFT/RIGHT/MIDDLE centro-parietal, LEFT/RIGHT/MIDDLE occipito-temporal regionsStatic imagesPositive, Negative valenceFace-in-the-crowd expression search task among neutral facesEarly gamma activity (100–200 ms) decreased with increasing detection difficulty; late gamma activity (after 400 ms) increased with increasing detection difficultySchubring & Harald, 2020 [121]EEGalpha, betaOccipitoparietal regionStatic images of scenesPositive, Negative valenceArousal levelBoth positive and negative valence pictures of high arousal levels associated with decreased alpha and lower beta band power; decrease in band power is associated with ERD response; a late ERS response observed for high-arousal negative picturesMartini  et al., 2012 [108]EEGP300, LPP, gammaLEFT/RIGHT temporal and frontal (negative valence), LEFT temporal and RIGHT parieto-temporal (neutral)Static images of scenesNegative valence, NeutralValenceIncrease in P300 and LPP components for negative valence across right hemisphere; enhanced gamma for negative valence; increased gamma power for negative valence with early low gamma and late gamma peaks across scalp over the whole duration; early strong between-region gamma synchronizations for negative valence between left and right temporal and frontal regions; late strong between-region gamma synchronization for neutral pictures between left temporal and right parieto-temporal regionsGüntekin & Basar, 2007 [117]EEGalpha, betaLEFT posterior areas (T5, P3, O2), LEFT frontal areas (F3, CZ, C3)Static imagesAngry, Happy, NeutralType of emotionOnly for pictures with highest valence ratings from subjects: higher amplitude of alpha power for angry than happy expression at mostly left posterior hemisphere; higher amplitude of beta power for angry than happy expression at mostly left frontal hemisphere


The alpha frequency band has also been widely studied in other research domains such as motor [122], attention [123,124], and working memory domains [125]. All of which have the common finding that, depending on the brain region(s) upon which it affects, alpha oscillations facilitate refining incoming inputs by temporally gating the activation of various neural populations via inhibitory mechanisms. This mechanism of function employed by alpha oscillation is therefore likely common for different processes in the brain, including emotion processing. Hence, it is not surprising that a similar inhibitory mechanism for alpha has been found by various groups of authors in the emotion recognition field.




**2. Beta Band**


If the alpha oscillatory response functions to ‘detect conflict’ by stepping up or down inhibition to filter out distraction/irrelevant sensory input or guide attention to relevant information, respectively, then the beta oscillatory response will function to ‘resolve conflict’. Studies have shown that it does so by maintaining equilibrium between incoming sensory processes and intrinsic cognitive processes [126,127,128]. With respect to emotion processes, the ‘conflict resolution’ characteristics of the beta frequency were observed in its enhanced responses to negative-valence emotion compared to positive ones [117,129,130]. Beta responses also seemed to be highly attuned to high arousal stimuli [131] where increases in beta responses occur for highly arousing images as compared to neutral ones.




**3. Theta Band**


The theta frequency band, extensively studied in animal models in working memory and memory, has been shown to be implicated in different types of emotion processes as well.

In the change detection of an emotion task, it was found that all modes of emotion changes resulted in enhanced theta synchronization, which correlates with the performance accuracy of subjects during emotion change detection, but the same cannot be concluded for alpha or beta bands [101]. Another study by Aktürk and team showed increased power and higher phase-locking values for theta (as well as all other frequencies) in the right parietal region when a face was first perceived and that this increase in power is greater for emotional expressions than neutral ones [132]. In addition, they found that theta phase-locking values were higher for fearful than happy or neutral expressions.

As seen in studies that compared theta responses to familiar versus unfamiliar faces, it was reported that theta activity was higher for familiar faces as opposed to the unfamiliar ones [133,134]. In terms of theta responses to valence-specific face expression, it appears that several studies report a stronger/higher theta synchronization for negative-valence emotion expressions [135,136,137] with the addition of heightened theta activity for emotion expressions of higher arousal levels [137].

The results gathered from investigations on theta oscillation corroborate moderately with the findings from the intracranial experiment done by Zheng and team in probing the functional relationship between the amygdala and hippocampus [119]. The experimental design was motivated by the degree of overlap between familiar and unfamiliar stimuli experienced by the subject. This feeling of ‘familiarity’, especially in ‘lure-correct-reject’ trials, would elicit an enhanced theta activity in the hippocampus as driven by a similarly enhanced theta activity in the amygdala due to decreased alpha power in the latter. Therefore, it is very probable that changes in global theta power and synchrony are modulated directly by amygdala activity as well.

Theta oscillation is also shown to be implicated in cognitive domains such as working memory [125,138,139], thereby implying that regions associated with working memory are likely recruited and modulated by the theta oscillation, e.g., the PFC. As such, one can postulate that since emotion processing and executive function are modulated by theta oscillation, detected and recognized emotions can be readily transmitted to the PFC for downstream decision-making. Subsequently, instructions can be transmitted to execute motor response(s) if an action is necessary in the event of an intense negative stimulus.




**4. Delta Band**


Delta oscillatory activity was demonstrated by several studies to exhibit a stronger synchronization for emotional stimuli as compared to that for neutral [136,140,141,142] in terms of arousal intensity. By virtue of the inverse relationship between oscillation and recruited neural network, a delta wave, being the slowest brain oscillation, is most likely going to involve brain-wide neural activity, which then implies that delta directly serves as the generic modulator for all cortical functions. From the brain’s perspective, this is advantageous because every neuron will possess membership in every functional network under the regulation of delta—making it fairly easy for the neuron to ‘switch membership’ whenever a particular input comes in. However, from a researcher’s standpoint, delta activity can be so widespread that it becomes difficult to acquire data that reflect finer/specific changes with an emotion interaction effect, e.g., valence. Further work on the delta band can be anticipated from the field in the near future.




**5. Gamma Band**


Gamma oscillations are the brain’s highest-frequency oscillations. They usually have very short temporal windows in which they occur, but they occur ubiquitously across the entire cortex. Due to the nature of the gamma oscillation, recruited neurons to be modulated by this high-frequency band are mostly small pools of localized populations. Since they are literally present everywhere on the scalp, researchers have been investigating them for correlational relationships to processes such as emotion recognition.

Martini and his group of authors investigated changes in gamma activity when subjects were present with images of emotional scenes. They found an enhanced gamma for negative valence images; with low gamma (30–45 Hz) and high gamma (65–80 Hz) peaks all over the cortex throughout the entire duration [108]. They also found strong between-region early gamma synchronizations for negative valence between left and right temporal and frontal regions, while strong between-region late gamma synchronizations were found for neutral valence between left temporal and right parieto-temporal regions.

In another study investigating low and high gamma bands via a ‘face-in-the-crowd’ expression visual search task, the authors found similar changes in both types of gamma bands: early gamma activity was reduced with increasing detection difficulty, while late gamma was enhanced with increasing detection difficulty [120]. Apart from that, the authors also found an effect of emotion valence on early and late gamma activity: the attenuation of low gamma activity with increasing detection difficulty only applies to positive valence face images but not for negative ones, suggesting that gamma activity in the early phase of emotion perception is likely guided by bottom-up attention features [143]. On the other hand, late gamma was not found to be affected by stimuli valence since both positive and negative emotion expression stimuli enhanced late gamma activity with increasing detection difficulty. It was then suggested that gamma activity in the later phase of emotion perception is likely guided by top-down attentional control.

Several other studies lend support to the finding that greater gamma responses are elicited by negative-valence face stimuli as compared to neutral or positive-valence ones [141,144,145].

Since the gamma frequency is the highest oscillating frequency in the brain, only neurons with cellular properties that can respond and keep up with such temporal dynamics are recruited [146,147]. As a result, the gamma frequencies recorded by EEG are mostly indicative of fast local neural computations that are transient in nature. As highlighted in several works above, gamma activity likely reflects the early sensory processing phase where it is mostly regulated by attentional mechanisms [98,139]. Therefore, it remains to be investigated if it truly holds robust information on emotions per se.

Given the plethora of information one can gather from evoked brain signals via EEG acquisitions, decomposing them into their respective frequency bands to observe band power changes, stimulus-dependent responses, amplitude changes, stimulus-evoked oscillation synchronization, and phase-locking values, respectively, would be ideal since neurons can be recruited and influenced by multiple oscillating frequencies at the same time.

This, however, will always remain a difficult task because the dynamic nature of the brain and its oscillations decree that pools of neurons are free to ‘switch memberships’ if there is a functional need for it to target a different downstream neuron. For example, removal of a stimulus from the attending field will cease the need for a subset of neurons to be placed “on standby mode” for activation. Hence, neurons and their populations are structured in a way where such resources can be readily reallocated to another functional network via ‘listening in’ and synchronizing with another oscillating rhythm.

One will have to bear in mind that every cell and every population in the brain are capable of initiating and maintaining a rhythm and that there are multiple oscillating bands happening simultaneously at any given instant. On a larger scale, these oscillating bands are constantly engaging and disengaging each other while maintaining their independence from some overarching global brain dynamics [148]. Consequently, each neuron is also constantly being recruited or dismissed from oscillating entities depending on the push–pull dynamics of multiple ongoing oscillations and their inherent cellular reactivity to in vivo factors such as the stimulus-driven/activity-dependent molecular neuromodulators circulating around the cortex. Additionally, one will also need to be aware that locally generated rhythms (e.g., gamma oscillations) can couple, decouple, or synchronize with each other via local network connections as well.

Now, imagine that on top of all that is already ongoing in the brain, an external stimulus is given to perturb the system…what exactly are we seeing on EEG readouts then?

The above depiction is intended to give readers, as accurately as possible, a snapshot of the dynamic workings of the brain at any given time point and hopefully allow the appreciation for the richness of information that is embedded in the very EEG signal readouts that we so conveniently acquire.

## 4. Affective Computing for EEG-Based Emotion Recongition

### 4.1. Experimental Protocols for Affective Computing

This section introduces the fundamental protocols used in EEG emotion recognition. These protocols, including publicly available datasets, data preparation techniques, and classification evaluation methods, are widely accepted and adopted in the field. They serve as essential benchmarks and guidelines for conducting EEG emotion recognition tasks.

#### 4.1.1. EEG Dataset

EEG datasets are essential for evaluating the average classification performance of models due to the complex interplay of variables involved in generating EEG signals. Unlike images or text, EEG signals are influenced by various intrinsic factors such as the subject’s cognitive state, race, sex, and age. Additionally, environmental factors, including equipment, electrode conductivity, and the nature of stimuli, further contribute to the variability in EEG signals. The DEAP dataset, from Table 4 below, widely utilized and referenced in research on EEG-based emotion analysis, was compiled and made accessible through collaborative efforts involving the Queen Mary University of London, the University of Geneva, and other institutions [149]. In this experiment, thirty-two participants watched a series of forty 60 s music movie clips while their EEG and peripheral physiological signals were recorded. During the viewing session, the participants self-evaluated their emotional experiences using a scale encompassing multiple dimensions such as Arousal, Valence, Like, Dominance, and Familiarity. These subjective ratings from the participants serve as emotional labels for the EEG samples and are used to optimize models in emotion recognition tasks. The MAHNOBHCI multi-modal dataset, designed for emotion detection and implicit tagging studies, is a benchmark dataset that captures various aspects of participants’ activities during emotion induction experiments [150]. It includes physiological and eye-tracking data, synchronized videos of participants’ faces and bodies, and audio recordings. The dataset contains data from 27 participants and encompasses EEG, video, audio, gaze, and physiological signals. Participants watched 24 videos designed to induce emotions and neutral videos during the data collection. In our work, our focus is specifically on analyzing the video content that participants watched and the corresponding EEG signals.

BCMI Research Center at Shanghai Jiaotong University released the SJTU Emotion EEG Dataset (SEED) [151]. The dataset includes 15 movie clips used to induce specific emotions, with three different emotional states represented. Each genre consists of five clips, each lasting approximately 4 min. All participants were Chinese and only selected Chinese movies to limit the potential influence of cultural background on emotional response. The dataset includes EEG data collected from 15 Chinese participants (seven males and eight females; mean age: 23.27 ± 2.37) over three different periods, allowing for the evaluation of algorithm robustness across varying data acquisition times. Subsequently, SEED-IV data [153] were released by SJTU, with most of the experimental designs similar to SEED. The experiment involved the use of a 62-channel ESI NeuroScan System and SMI eye-tracking glasses to capture the signals of 15 subjects for three sessions, each containing 24 trials. The dataset included 72 film clips categorized into four emotions (neutral, sad, fear, and happy), and the subject watched six film clips per emotion class in each session. SJTU released another SEED-V dataset [156], which increased the data collection to 20 subjects (10 males and ten females) who were students at SJTU. In SEED-V, there was a 15 s hint of starting before the elicitation phase before ending with 15 or 30 s of self-evaluation. Negative stimuli such as disgust or fear were given 30 s for evaluation, while happiness, neutrality, and sadness were given 15 s. The scoring range for each emotion was between 0 and 5, with 5 representing the best induction effect and 0 the worst (e.g., do not feel anything). If the subject’s mood fluctuated during watching, it was also given a 0 score. The natural state was 5 points. The subjects watched 15 stimuli containing 3 of each emotion (happy, sad, neutral, fear, and disgust). The equipment used comprised the 62-channel ESI NeuroScan System and SMI eye-tracking glasses.

The Dreamer dataset includes simultaneous recordings of EEG and ECG signals during audio-visual emotion elicitation experiments [152]. The experiment was conducted on twenty-three participants who performed valence, arousal, and dominance self-assessments after each trial. Wireless and cost-effective equipment were used to create the dataset, highlighting its scalability to a recognition approach in practical applications. The Multi-modal Physiological Emotion Database (MPED) [154] encompasses multiple modalities, including 62-channel EEG, ECG, respiration, and galvanic skin responses, recorded from 23 Chinese student volunteers. The dataset features a selection of 28 Chinese video clips, derived from a pool of 1500 clips encompassing film clips, TV news, and TV shows, representing seven discrete emotions (joy, funny, anger, fear, disgust, sadness, and neutrality). The experiment consisted of two sessions, spaced at least 24 h apart, during which each participant watched 14 video clips per session, resulting in a total of 14 trials.

#### 4.1.2. Data Preparation

Data preparation has a significant impact on the recognition performance, especially in the domain of EEG emotion recognition. Currently, two main approaches for preparing data for validation are subject-dependent and subject-independent. The former involves training and testing the same subject’s data to evaluate intra-subject variabilities, while the latter aims to build a generalized model for different subjects or data domains to allow the evaluation of unseen subjects, thereby measuring inter-subject variabilities. In both approaches, K-fold cross-validation is commonly applied for an overall evaluation, where the dataset is split into K-independent parts for training and testing. For the subject-dependent approach, usually, unseen trials are left out; this splitting method is called leave-K-trial-out. The subject-independent approach is known as the leave-K-subject-out, and a widely adopted approach is the leave-one-subject-out (LOSO) for validation. In the subject-dependent experiment, individual subjects’ data are trained and validated, and the mean across the subjects is considered the average classification performance for the entire subject-dependent approach. However, the limitation of the this approach is the tendency to overfit the model due to a limited dataset size, which inevitably gives rise to skew performance.

In considering most of the recognition approaches in recent years, segmentation to the EEG trial is applied. There are pros to this approach, as firstly, it increases the dataset, and secondly, it helps to improve the performance with shorter segments. However, the drawback of segmenting is that it begets the question of how long the EEG segment must be for registering the evoked emotion on the EEG signal and whether time lags on the cognitive elicitation affect the signal. Nonetheless, segmenting helps reduce redundant data features and increases the data quantity, allowing the deep learning models to perform better.

Such splitting and segmenting of data can lead to a subject-bias condition where information from the training dataset enters the validation dataset. Consequently, a high-recognition performance will entail, and this is an erroneous approach for EEG emotion classification. This subject-bias situation occurs when the random assignment of trial segments to training and validation datasets comes from the same trial. In comparing between subject-dependent and subject-independent approaches, the latter avoids this subject-bias situation naturally due to using unseen subjects’ EEG data.

#### 4.1.3. Evaluation of Classification

The evaluation of recognition models in emotion recognition tasks requires comparing predicted emotion ratings/labels with the ground truth of the benchmark datasets. In most situations, EEG-collected data are small, and with the advent of deep learning approaches, these small datasets can lead to class imbalance. Thus, while an accuracy approach might be sufficient to determine the performance of the recognition model, an F1 score might be more appropriate to handle such imbalances.

A limitation of the ground truth is that it can either be a self-assessed emotion label or be assessed by other observers of the stimuli. For the former, it will be closer to the experience of the subject’s signal semantic, while the latter will be less accurate in depicting the subject’s emotional state. Nonetheless, most available datasets will assume a universal emotion for the particular stimuli, which can become a significant limitation of the experiment. Additionally, the average observer’s annotation of the ground truth might lead to a noisy label for the EEG subjects. Nonetheless, supposedly happy stimuli can reflect as a sad elicitation in some subjects, in some likely situations due to their past trauma. Ultimately, for EEG emotion recognition, the best labels should not be aggregated but unique for each subject associated with the particular stimuli.

### 4.2. Feature Engineering and Preprocessing Techniques

Feature engineering and preprocessing are crucial for learning representations from EEG signals. They can be a determining factor for consideration of whether the learning model can perform well during validation. This chapter explores the cases where feature engineering and preprocessing are designed based on the spatial dimensions of the cerebral regions of the brain. This chapter aims to understand how feature engineering and preprocessing lead to better performance of the models by providing a neuroscience perspective for analysis.

Adopting a classification of features approach [157], this review extends this classification of methodologies regarding a neuroscience perspective and interpretations. This chapter uncovers the computational approach of feature engineering in terms of the spatial aspect of neuroscience.

#### 4.2.1. Feature Processing

Feature preprocessing for EEG signals is important for providing the salient features for the recognition task by removing noise and artifacts. The human body does not generate neural signals due to emotions, electrical noise, and technical issues, such as the conductivity of the gel, are the main contributing factors to noise and artifacts. For example, the EOG artifact at a frequency less than 4 Hz due to eye blinking, ECG artifacts at 1.2 Hz, and EMG artifacts at more than 30 Hz are some of the noises created by the human body. In addition, electrical noise and environmental noise often manifest at a frequency above 50 Hz. Doing so, however, will inevitably remove crucial high gamma (>50–80 Hz) signals, which can potentially be useful towards subsequent emotion recognition and classification—a point which can be put up for discussion in a later section. Nonetheless, in such instances, the usual first preprocessing performed on the dataset will be applying band-pass filters to remove most of these artifacts. Alternative methods such as independent component analysis (ICA) and discrete wavelet transform are applicable for removing noise due to specific frequency bands.

Removing noise and artifacts reflects the preservation of signal that pertains only to the neural signals originating from the brain. As a result, the preprocessed signals will consist mainly of the brain’s neural signals, allowing a better representative signal of the evoked emotion. An empirical approach to performing this preprocessing will be subtracting the reference electrode from the measurement electrodes. This approach helps to reduce the noise because the reference electrode acts as a control that captures none of the brain neural electric potentials while capturing the environmental interference the electrode will record. Experimentally, this reference electrode is often located at the subject’s body or close to the head region, such as the earlobe, nose, mastoid process, and collarbone. However, when such reference electrodes are not found, the average of all signals can be used as a reference signal for subtraction from the measurement electrodes.

While not exhaustive, these preprocessing approaches mainly consider the spatial location of the brain where signals are being measured. Removing the noise and artifacts improves the signal-to-noise ratio, thereby offering a more explicit resolution of the brain signals in response to the emotion elicitation. With a better resolution, these signals provide better representative features for the classification models [158].

#### 4.2.2. Feature Engineering: Time Domain

Time domain-engineered features have been widely used to study brain function since EEG acquisition is based on measuring electrical potentials along the time dimension. Common methods adopted for time-domain analysis include statistical features (i.e., mean, variance skewness, kurtosis, etc.) [24], Hjorth parameter [159], higher-order crossing (HOC) [160,161], and event-related potentials (ERP), which are designed to bring out the representative temporal information from the EEG signal. For instance, the modified Hjorth parameters [162] for electrodermal signals create features based on the descriptors, namely, activity, mobility, complexity, chaos, and hazard, which can be used to classify arousal and valence. This time domain analysis preserves the temporal information in EEG signals but is complex due to the diverse waveform patterns. Consequently, a standardized approach to analyzing EEG time-domain features is difficult to define. Thus, specific analysis can be more applicable in some scenarios than others.

For instance, the correlation between specific components of ERP and emotional states has been defined in a way in which the early ERP components correlate with valence [163]. In contrast, late ERP components are associated with arousal [164]. Thus, the findings suggest that different ERP components can provide insights into various aspects of emotions. Examples of ERP components studied include P100, N100, N200, P200, and P300. Despite such empirical evidence, they are not practical for real-time or online applications as they are typically obtained by averaging multiple EEG trials [157]. Furthermore, in view of recent works, had these been effectively useful for emotion recognition, the current trends in EEG emotion recognition models will have adopted ERP widely. Since these time domain features reflect the brain’s electrical activity associated with cognitive processes over time, emotional responses, and information processing, they become reliable representations for learning models to train on these features.

#### 4.2.3. Feature Engineering: Frequency Domain

Frequency domain analysis methods provide robustness to noise and a different dimension to observe the signal by converting EEG signals from the time domain to the frequency domain. The most common feature adopted in emotion classification is the power spectral density (PSD), which offers the observation of the measured signal power with respect to the specific frequency as well as a quantitative measurement of the power relationship between the frequencies in question. Then, given a frequency range, the signals can form frequency bands. This transformation to the frequency dimension is done by applying a fast Fourier transform (FFT) to the raw EEG signal. The resultant frequency range can further be decomposed into distinct frequency bands such as Delta, Theta, Alpha, Beta, and Gamma. However, due to the nature of the non-stationarity EEG signal, neural signals are often unevenly time-measured, and each timestep will contain different shapes of periodic signals. Hence, the Welch approach is introduced, breaking down the signal into segments and using the Hanning window to generate the PSD. Variants such as the logarithm energy spectrum and Differential Entropy (D.E.) are proposed to improve the PSD approach. The latter variant, D.E., measures the entropy for continuous random variables based on the probability density function. A probability density function based on this logarithm PSD is defined by approximating these PSD into the logarithm energy spectrum for a specific frequency band of fixed-length EEG sequence. The analytical approximation [25] for differential entropy of the PSD is thus defined as the logarithm function of the PSD with a constant for a specific frequency band:(1)$$\begin{array}{c} DE\left(P_{band}\right)=h_{band}=\frac{1}{2}\log2\pi \exp\sigma_{band}^{2}\\ =\frac{1}{2}\log P_{band}+\frac{1}{2}\log\frac{2\pi \exp}{N} \end{array}$$
Alternative features such as a higher-order spectrum (HOS) can be, too, classified under the frequency domain analysis.

From a neuroscience viewpoint, different frequency bands are associated with specific neural processes and cognitive functions that are often stimulus-dependent. One dominating oscillation band or a combination of multiple bands can be recruited at stimulus/task onset. Nonetheless, the Delta band is related to deep sleep, while the Gamma band is linked to attention and sensory processing in general. By analyzing the power distribution, it is possible to identify patterns and changes in brain activity related to a given stimulus and in the case of an emotional stimulus to emotional processes.

#### 4.2.4. Feature Engineering: Time–Frequency Domain

The combination of time and frequency domains can capture the dynamic characteristics of non-stationary signals like EEG. The short-time Fourier transform (STFT)) [26] is one of the dynamic time–frequency joint analyses performed by applying a fixed window function to the time domain. This transformation treats the non-stationary signals as a combination of short-time stationary signals. Alternative methods such as wavelet transform, wavelet packet transform (WPT), and Hilber-Huang transform (HHT) are also commonly used in time–frequency analysis, providing higher resolution and analysis for time-varying and non-stationary signals. The difference between them is that the former two are based on the wavelet approach with the advantage of observing the spectral changes in power over time, while the latter, HHT, is based on empirical mode decomposition that extracts oscillatory-like features from the data. EMD has a distinct advantage over wavelet convolution methods as it enables the detection of subtle frequency variations [165].

In EEG-based emotion recognition, wavelet features have been more widely adopted in recent years [166,167], such as the discrete wavelet transform (DWT) [168], and continuous wavelet transform (CWT) [23]. For instance, the DWT provides approximation and detail coefficients at various scales, and the process can be iterated to obtain coefficients at even finer scales. The wavelet entropy and wavelet energy can be computed for each frequency range using these coefficients. Given the EEG signal measurement, the wavelet mother and decomposition levels yield different frequency ranges.

Other variants of EMD include multivariate empirical mode decomposition (MEMD) [169] and variational mode decomposition (VMD) [170]. VMD can be extended to a multivariate variational mode decomposition (MVMD) [171] to develop a time-frequency analysis of multivariate nonstationary signals to extract P-predefined multivariate modulated oscillations. These techniques generally decompose multi-channel EEG signals into multiple intrinsic mode functions (IMFs) to extract more representative features.

Stockwell transformation features combine the advantages while compensating for the weaknesses of STFT and wavelet-based transformation by considering a modified Gaussian window function [172]. Another combination technique uses variable-frequency complex demodulation (VFCDM) [173] to prepare high-resolution biomedical signals. VFCDM is achieved in a two-step process. First, complex demodulation estimates the time-frequency spectrum of the dominant frequency band of interest. The final step involves converting the signal into a series of band-limited using a collection of low-pass filters before applying the Hilbert transform to obtain the analytical signal. Despite the current efforts in VFCDM with the CNN approach applied in epilepsy detection, the results yielded from this feature processing method suggest a feasible feature engineering approach for EEG-based emotion recognition.

Time–frequency analysis helps analyze non-stationary signals where the frequency components change over time. The dynamic changes in neural activity associated with cognitive and emotional processes can be captured by decomposing the signal into different scales and frequencies.

#### 4.2.5. Feature Engineering: Nonlinear Domain

The brain’s neural system activities have been identified to exhibit nonlinear signals. Considering nonlinear neural signal properties, it would be natural to apply nonlinear dynamic methods to make sense of these neural signals. Notably, EEG measurements are found to be multi-fractal dimensions [174], and different cognitive tasks elicit distinct brain states that align with specific fractal dimensions [157]. However, these features have high computation costs and are sensitive to parameter settings. Furthermore, with the popularity of deep learning-based models, some of these features might fall in favor of others due to the computation cost.

While differential entropy is a log-based function, which can be considered a nonlinear feature, we believe it is more closely associated with PSD features. Hence, we classify these DE features as time domain features. For more details and another perspective on these features, the review papers of Li et al. [157] and Garcia-Martinez et al. [175] can offer these insights.

Veeranki et al. [176] proposed multiple non-linear methods for emotional recognition based on electrodermal activity, such as isaxEDA [177]—capturing temporal patterns and fluctuations to capture dynamic short-term fluctuations, comEDA [178]—extracting features that reflect the level of recurrence and predictability, netEDA [179]—establishes connections between each data point based on similarity to capture relational structure, and topEDA [180]—examines the intricate patterns and shapes of the geometric properties of the plane boundary of the signal. Each feature allows unique patterns to discriminate temporal and spatial patterns, network structures, and variations in emotional arousal. As a result, a combination of these approaches could benefit emotion recognition with EEG signals.

One could argue that nonlinear features work in EEG emotion recognition because the human body comprises nonlinear dynamical systems, such as the brain and heart. In addition, the brain itself is a convolution of intrinsic noise composed of microscopic noise coming from changes in a single neuron’s membrane potential in the form of EPSPs (excitatory post-synaptic potentials) and IPSPs (inhibitory post-synaptic potentials) to macroscopic noise stemming from the ongoing coupling, decoupling, synchronizing pools of neurons. Thus, the neurophysiological signals acquired in the form of scalp EEG readouts exhibit chaotic characteristics [157]. These nonlinear features allow for studying the brain’s nonlinear dynamics, such as understanding the underlying mechanisms of cognitive processes, emotion regulations, and brain disorders.

#### 4.2.6. Feature Engineering: Spatial Dynamics Features

In the earlier review by Li [157], there are two features, asymmetry features and brain network features, that are highlighted in addition to time, frequency, or time–frequency domain features. However, both of these features share a common trait based on the measurements of the different regions of the brain. Hence, in this review, both of these features can be grouped under the classification of spatial dynamics features. Nonetheless, this section will provide an independent review of the abovementioned features for a more distinctive illustration of these differences. More importantly, a deeper analysis will reveal that most of these proposed features from asymmetry and brain network features can be further reclassified under the domains of time, frequency, and time–frequency.




*Asymmetry Features*


Asymmetry features encompass the neural signals that exhibit non-symmetrical patterns between the left and right hemispheres of the brain. These signals hold valuable information about cognitive processes occurring within the brain. The brain lateralization phenomena in emotion processing and mood disorders have been observed through brain imaging studies [181,182]. The inherent structural and functional differences between brain hemispheres give rise to differences in EEG characteristics between symmetrical brain positions, such as spectral power asymmetry (SPA). The asymmetry feature is obtained by calculating the difference or ratio of indexes from different signal sources [183]. Extending the DE features to asymmetry features, differential asymmetry (DASM) and rational asymmetry (RASM) features have also been proposed. The former is the difference, and the latter is the ratio between the DE features of the pairs of hemispheric asymmetry electrodes [151,184].

Since the brain exhibits lateralization phenomena in a different location during emotion processing, these asymmetry features can contain information on functional specialization and interhemispheric communication in emotional and cognitive processes.




*Brain Network Features*


Brain network features from various brain regions are intricately linked and play a crucial role in high-level cognitive processes and emotions. The EEG signals measured from different brain regions can capture these inherent connections and provide information about the dynamics of the brain. Brain networks are constructed by studying time correlations or spectral coherence between multi-channel brain signals using various techniques such as Pearson correlation coefficients, phase-locking value (PLV) [185], Granger causality [186,187], and more. While the list is not exhaustive in this review, these techniques allow researchers to estimate the interactions and relationships between different brain areas.

The abovementioned brain network features involve handcrafted features based on the domain knowledge of the dynamics of the brain and how emotions are evoked in humans. This approach becomes impractical as most of the features are based on time-series analysis methodologies, given that the correlation with emotional states and the extent of synchrony are mainly unknown. Furthermore, physiological and cognitive factors can introduce variations in EEG signals, potentially impacting the reliability and effectiveness of these handcrafted features.

With the advent of deep learning approaches, incorporating graph theory in these models will allow the spatial channels of the EEG electrodes to simulate each node on the graph [188]. And using these graph-based deep learning models allows the model to learn the features based on the designed optimization of the model. Consequently, it allows the study to circumvent the need to design handcrafted features. These models will automatically learn valuable insights into the brain network, even if the initial input signal comes from raw signals, PSD, or DE features.

Mapping brain network features can offer insights into how emotional states are processed and regulated within the brain. Notably, the graph-based models allow the model to automatically unravel the complex interactions and information flow within the brain by learning and generating these brain network features, which then can be used for downstream tasks such as emotion recognition.

### 4.3. Recognition Models and Algorithm

With an understanding of feature processing, it would be imperative that the model could process this information to recognize the associated emotion. While some features are more popular than others for emotion recognition, the effectiveness of these features is needed for the recognition task. In this section, instead of reviewing the models used for EEG emotion recognition and elucidating the different steps or pathways the recognition models will undertake, this section reviews these models from a neuroscience perspective and evaluates the effectiveness of these approaches.

In particular, this review identifies four categories based on the neuroscience perspectives for EEG emotion recognition tasks. Specifically, spatial, temporal, spatiotemporal, and generalized modeling are the categories this review will be associating with the recognition models and their methodology. These categories are associated with directly modeling the brain elicitation of emotions rather than associating with the features’ spatial and temporal dimensions. This section reviews the effectiveness of these recognition models constructed to unravel the emotional information from EEG signals from a neuroscience perspective. It allows us to understand the necessity of modeling the brain for recognition models and identify the first principles that make EEG emotion recognition successful. We have designated machine learning and deep learning categories within each subsection, whichever is applicable. The former refers specifically to non-deep learning model methodologies, while the latter utilizes the deep learning model as an entire or part of the recognition model.

#### 4.3.1. Spatial Dynamics Recognition Models

From a neuroscience perspective, the spatial dynamics of emotion are defined as the brain’s neural structures that provide the modulatory effects in response to emotional stimuli. Arguably, the segment of an EEG trial is often used as a feature, which inherently contains temporal information. In these models, the features based on a short EEG trial segment represent static neural information.




*Machine Learning Approach*


Miranda-Correa et al. [155] introduced the AMIGOS dataset. Specifically, the authors use PSD for feature extraction in the EEG-only recognition model before using the Gaussian Naïve Bayes classifier to recognize emotions.




*Deep Learning Approach*


In some recognition models, the handcrafted features use frequency domain analysis, such as DE and PSD, which can be considered the temporal aspect of brain dynamics. However, these features tend to be based on the short EEG segment, which allows us to regard this as a stationary neural signal capture at a particular moment. For instance, Song et al. [189] proposed a model called dynamical graph convolutional neural networks (DGCNN), considering the dynamic nature of functional networks—their work extracts DE, PSD, DASM, and RASM from the five different frequency bands. And the learning model learns by extracting discriminative characteristic and functional connectivity information. DGCNN is an improved version of the GCNN model based on EEG-derived graph networks [190], built from the fusion of the within-frequency functional connectivity graph (FCG) and the cross-frequency FCG. In contrast, Wang et al. [191] proposed the BDGLS, which improves GCNN by introducing a broad learning system.

In contrast to traditional GCNNs, the DGCNN’s adjacency matrix is not static but dynamically updated during training. The aim is for the learned adjacency matrix to capture the inherent spatial correlation among the EEG channels with the vertex representation of the graph comprising the handcrafted features. This adaptive nature of the adjacency matrix simulates the everchanging brain dynamics allowing it to outperform the GCNN-based approaches.

Subsequently, Zhong et al. [188] proposed a regularized graph neural network (RGNN) to leverage the EEG channel topology, aiming to capture the spatial dynamics of the brain through the adjacency matrix in the graph neural network. This graph network enables the learning of local and global relationships among EEG channels. The model incorporates two regularization techniques: node-wise domain adversarial training (NodeDAT) and emotion-aware distribution learning (EmotionDL). NodeDAT is a regularization mechanism to address domain shifts in each channel, allowing the model to generalize to inter-subject recognition scenarios effectively. On the other hand, EmotionDL enhances recognition performance by learning to predict emotion via label distribution instead of discrete labels.

#### 4.3.2. Temporal Dynamics Recognition Models

The term “temporal dynamics” in this section refers to recognition models that capture the time-specific aspects of brain-based emotion. These temporal patterns are distinctive neural signatures of the time-specific aspect as they unfold sequentially across specific time windows and brain structures. Models falling into this category primarily focus on modeling the temporal dynamics of the brain. In this section, the distinguishing factor of a temporal dynamics recognition model is its consideration of neural signature changes, specifically between different brain structures within a given time window.




*Deep Learning Approach*


One of the earlier works to explore deep learning in EEG emotion recognition was conducted by Thammasan et al. [192]. They proposed using a deep belief network (DBN) for classifying EEG emotions. Their model employed three handcrafted features—FD, PSD, and DWT—based on a segmented length of 1–4 or 5–8 s and utilized DBN for the classification task. Although the learning model does not rely on brain emotion elicitation dynamics, this work falls under the category of temporal dynamics models due to the utilization of the abovementioned handcrafted features. Observing the effects of deep learning models learning directly from EEG signals is interesting. Hu et al. [193] showcase with ScalingNet that using deep models can extract spectrogram-like features for downstream classification.

#### 4.3.3. Spatiotemporal Dynamics Recognition Models

Referring to Section 3.2, spatiotemporal dynamics recognition models refer to the class of models that associate with the spatial and temporal regions of the brain dynamics. Notably, the regions of the brain that modulate the emotional experience and the temporal mechanism of brain structure work in tandem to modulate particular emotional experiences.




*Deep Learning Approach*


To enable long-term emotion monitoring, Li [23] Table 5 proposes a C-RNN approach that deviates from the existing approaches that focus on segment-level emotion recognition tasks, where emotions are recognized for signal segments lasting 1 s or more. Instead, it adopts a continuous sampling and prediction approach, where each segment is predicted individually, and the frame with the highest probability in the entire trial is selected as the predicted emotion label. In this work, feature engineering considers that the CWT is followed by scalogram analysis to generate features that capture the spatial relationships among multiple channels. Then, the mode C-RNN will generate features with CNN before the LSTM sample features and, based on prior and current features, predict that frame segment. This learning model thus attempts to learn the temporal changes, while feature engineering generates the spatial relationships. As a result, this spatiotemporal approach that uses spatial feature engineering and temporal model learning helps improve recognition performance.

Using RNN models such as LSTM to track and learn the temporal dynamics of the EEG signal, e.g., C-RNN, can be argued to model the temporal brain regulation process. Yin et al. [194] proposed ECLGCNN, which uses the GCNN and LSTM for emotion recognition. The feature processing involves calibrating EEG data, extracting the differential entropy from each segment, and concatenating these features to form a feature cube. GCNN learns the spatial information of channel connectivity for a segment of the EEG signal, and the LSTM learns the evolution of the channel connectivity and predicts the emotion similar to C-RNN.

For a graph-based model that models the spatiotemporal dynamics of the brain, Li et al. proposed STGATE, a transformer-encoder with spatial–temporal graph attention that uses DE features from EEG signals for EEG emotion recognition [27]. It introduces the transformer learning block to learn the electrode-level time–frequency from EEG signals before having the spatial–temporal graph attention to capture the correlation between EEG electrodes in the spatial domain and temporal information of the signal. Additionally, Li et al. showed that utilizing this raw signal enables a direct mapping and visualization of the adjacency matrices through a topographic map, thereby emphasizing the significance of the frontal lobe in emotion recognition. However, upon examining the ablation study of STGATE, a crucial piece of information was revealed: 2D convolutions may play a vital role in achieving good performance in the modeling process. It can be inferred that both the baseline model, constructed with a series of multi-kernel 2D convolutions, and the transformer learning blocks contain the 2D CNN modules. Naively, from the neuroscience perspective, it can be argued that these CNN, when trained on the raw signals, inherently learn to analyze both the spatial mapping of the brain regions (represented by the channels) and the temporal information of the signal (defined by the EEG signal segments).

Instead of relying on handcrafted signals, novel models have been introduced to learn directly from the EEG signals. Lew et al. [28] proposed a regionally operated domain adversarial network (RODAN) that incorporates a group of GRU models and an attention mechanism, allowing for the consideration of spatial–temporal relationships among brain regions and across time. Their work also incorporates a domain adversarial network to mitigate the distribution shift between training and testing data. This learning model effectively captures the spatiotemporal dynamics of the brain through RODAN and demonstrates the case of subject-bias experiment boosting classification metrics. From a neuroscience perspective, this can be attributed to the consistent neural patterns exhibited during emotion regulation within the same trial. Thus, when a recognition model is trained using subject-bias experiments, it effectively learns to identify the patterns present in the validating dataset, even if it was only trained on the training dataset.

Interestingly, Song et al. [195] introduce GECNN to classify the SEED, SDEA, Dreamer, and MPED datasets by introducing recognition from constructed EEG-based images generated based on converting the EEG features (i.e., DE and PSD) into EEG images.

#### 4.3.4. Generalized Recognition Models

While it may not be practical to categorize every recognition model according to previous frameworks, it is essential to acknowledge the existence of generalized recognition models that do not model brain dynamics. These models are still valuable, as they enable emotion recognition using EEG data. Considering the generalized recognition model, both engineered features and the learning model must not model brain dynamics.




*Machine Learning Approach*


In a non-deep learning approach, Atkinson and Campos [24] demonstrated that with the feature selection method based on minimum redundancy maximum relevance (MRMR) and the SVM with RBF classifier, the computational approach does model the way a brain elicits emotion. While it does not model brain dynamics, the MRMR approach selects features that associate best for classification, thus producing salient features for improving classification. However, this approach does not elucidate brain dynamics and solely relies on proven computations that can identify and differentiate patterns. Ackermann et al. [196] introduce multiple handcrafted features based on the selection using MRMR on extracted features of statistical, Hilbert–Huang spectrum (HHS), STFT, and HOC from each frequency band before using random forest or SVM to classify the features into emotions. In their work, the EEG segment used is based on a 1 s window (which can be assumed as static or stationary data), and with their learning model approach, it can be inferred that there is no temporal or spatial learning of brain dynamics involved.
brainsci-14-00364-t005_Table 5Table 5Recent works on EEG emotion recognition.AuthorDatasetMethodologyExperiment ResultsNeuroscience AttributeAtkinson and Campos, 2016 [24]DEAPFeature selection: Minimum-Redundancy-Maximum-Relevance (MRMR), Model: SVM classifier with RBF KernelSubject-Independent: DEAP: 2-Class (Valence-73.14%, Arousal-73.06%)GeneralizedThammasan et al., 2016 [192]EEGs induced by MIDI audio materialsFeature Processing: Fractal dimension (FD), power spectral density (PSD), discrete wavelet transform (DWT); windowing size (1-4/5-8 secs), Model: Deep belief network (DBN)Subject-Dependent: 2-Class (Valence-88.24%, Arousal-82.42%)TemporalLi et al., 2016 [23]DEAPFeature Processing: CWT and Scalogram on multi-channels, Model: C-RNN; CNN for extracting features and LSTM to predict the emotion2 classes (Valence: 0.7206, Arousal: 0.7412)SpatiotemporalYin et al., 2017 [197]DEAPFeature Processing: Statistical Features and PSD Model: Multi-Fusion-Layer based Ensemble Classifier of Stacked Autoencoder (MESAE)Subject-Dependent: {(2-Class: (Valence-72.43%, Arousal-69.01%))}SpatialLi et al., 2018 [198]SEED, SEED-IVFeature Processing: DE; Model: Deep Adaptation Network (DAN) using MK-MMD lossesSubject-Independent: {(SEED: (DAN-83.81%, DANN-79.19%)), (SEED-IV: (DAN-58.87%, DANN-54.63%)}SpatialYang et al., 2018 [199]DEAPFeature Processing: Raw EEG signal built into data mesh based on electrode topology; Model: Parallel Convolutional Recurrent Neural NetworkSubject-Dependent: {(DEAP: 2-Class (Valence-90.80%, Arousal-91.03%))}SpatiotemporalZhang et al., 2018 [200]SEEDFeature Processing: DE, Model: Spatial Temporal RNN (STRNN) to learn spatial and temporal dependenciesSubject-Independent: {89.5%}SpatiotemporalZheng et al., 2018 [153]SEED-IVFeature Processing: Differential Entropy (DE); Model: EmotionMeterSubject-Dependent: {SEED-IV: 70.58%}SpatialWang et al., 2018 [191]SEEDFeature Processing: Differential Entropy (DE); Model: BDGLSSubject-Dependent: {SEED: 93.66%}SpatialLi et al., 2018 [201]SEED, SEED-IVFeature Processing: DE; Model: Bi-hemispheres domain adversarial neural network (BiDANN)Subject-Dependent: {92.38%}, Subject-Independent: {83.28%}SpatiotemporalLi et al., 2019 [202]SEEDFeature Processing: DE; Model: R2G-STNN, based on BiLSTM to learn the spatial and temporal featuresSubject-Dependent: {93.38%}, Subject-Independent: {84.16%}SpatiotemporalZhong et al., 2020 [188]SEEDFeature Processing: Differential Entropy (DE) smoothed by linear dynamic systems(LDS), Model: Regularized GNN (RGNN) with Emotion DLSubject-Dependent: (SEED: 92.24%), (SEED-IV: 79.37%), Subject-independent modeling: (SEED: 85.30%), (SEED-IV: 73.84%)SpatialSong et al., 2020 [189]SEED, DREAMERFeature Processing: Using handcrafted features such as DE, PSD, DASM, RASM and DCAU based on frequency bands, Model: Dynamical graph convolutional neural network (DGCNN)subject-dependent modeling (SEED: 90.4%, DREAMER-Valence: 86.23%, DREAMER-Arousal: 84.54%), subject-independent modeling (SEED: 79.95%)SpatialLew et al., 2020 [28]DEAP, SEED-IVFeature Processing: RODAN extracted features; Model: RODAN, A spatial–temporal model with a domain adversarial network.Subject-Dependent: {DEAP: 2-Class (Valence: 62.93%, Arousal: 63.97%), 4-Class (VA: 38.16%); SEED-IV: 70.28%}, Subject-Independent: {DEAP: 2-Class (Valence: 56.78%, Arousal: 56.60%), 4-Class (VA: 31.84%); SEED-IV: 60.75%}SpatiotemporalDuan et al., 2020 [203]DEAPFeature Processing: None. No handcrafted features; Model: CNN with Meta Update StrategySubject-Independent: {(DEAP:2 Class(Arousal: 66.5% )}GeneralizedDuan et al., 2020 [204]SEED, DEAPFeature Processing: No handcrafted Features. Features generated from CNN network; Models: MLCL that adopts meta learning approachSubject-Independent: {(DEAP: 2-Class(Arousal 67.5%)), SEED: 3-Class( 78.6%))}SpatiotemporalLi et al., 2020 [205]SEED, SEED-IV, MPEDFeature Processing: DE for SEED and SEED-IV, STFT for MPED, Model: Bi-hemispheric discrepancy model(BiHDM) capture the asymmetric characteristics between hemispheresSubject-Dependent: {(SEED: 93.12%), (SEED-IV: 74.35%), ((MPED:40.34%) }, Subject-Independent: {(SEED: 85.4%, SEED-IV: 69.3%, (MPED: 28.27%)}SpatiotemporalDing et al., 2021 [206]SEEDFeature Processing: DE; Model: Task-specific Domain Adversarial Neural Network (T-DANN) an adversarial training method that transfer information inter- and intra-subject for predictionSubject-Independent: { (cross-subject: 74.19%, cross-phase: 85.13%)}SpatialWang et al., 2021 [207]SEED, DEAPFeature Processing: DE; Model: Few-label adversarial domain adaption (FLADA)Subject-Independent: {(DEAP:68.0%), (SEED: 89.32%)}SpatialMiranda-Correa et al., 2021 [155]AMIGOSFeature Processing: PSD, PCA; Model: Gaussian Naïve Bayes ClassifierSubject-Independent: {(AMIGOS: 2Class(Valence-56.4%, Arousal 57.7%)}SpatialLiu et al., 2021 [156]SEED, SEED-IV, SEED-V, DREAMER, DEAPFeature Processing: Differential Entropy (DE); Model: DCCASubject-Dependent:{(SEED: 94.6%),(SEED-IV: 87.5% ), (SEED-V: 85.3%), (DREAMER: Valence-90.6%, Arousal-89.0%, Dominance-90.7%), (DEAP-Valence: 85.6%, DEAP-Arousal:84.3%)}SpatialYin et al., 2021 [194]DEAPFeature Processing: DE Features, Model: GCNN-LSTM hybrid model, named ECLGCNN.2 classes, subject-dependent modeling (Valence: 90.45%, Arousal: 90.60%), subject-independent modeling (Valence: 84.81%, Arousal: 85.27%)SpatiotemporalHu et al., 2021 [193]DEAP, AMIGOSFeature Processing: None, use Scaling Layer to generate spectrogram-like features Model: ScalingNetSubject Independent: {(DEAP: 2-Class (Valence-71.88%, Arousal-71.8%, Dominance 73.67%))}TemporalSong et al., 2021 [195]SEED, SDEA, DREAMER, MPEDFeature Processing: DE and PSD for SEED, PSD and HHS for SDEA, PSD for DREAMER and MPED; features converted into EEG-based images. Model: GECNN, graph-embedded convolutional neural network.Subject-Dependent: {(SEED: 92.93% ), (SDEA: 79.69), (DREAMER: Valence-95.73%, Arousal-92.79%)), (MPED: 40.98%)} Subject-Independent: {(SEED: 92.93), ( SDEA: 53.31%}SpatiotemporalLiu et al., 2022 [208]SEEDFeature Processing: No handcrafted Features, raw signal. Model: 3DCANN, 3D-CNN layers for 5 × 1 s raw EEG signals, then attention block and classification.Subject-Dependent: {(SEED: 97.35%)}, Subject-Independent: {(SEED: 96.37%)}SpatiotemporalLiu et al., 2023 [209]SEED, SEED-IV, DEAPFeature Processing: DE for SEED and SEED-IV; Model: EeTSubject-Dependent: {(DEAP:2-Class (Valence-92.86%, Arousal-93.34%)), (SEED: 96.28% ), (SEED-IV: 83.27%)}SpatiotemporalLi et al., 2023 [27]SEED, DreamerFeature Processing: DE feature; Model: STGATE, a transformer-encoder is applied for capturing time-frequency features and graph attentionSubject-Independent: {(SEED: 90.37%); (SEED-IV: 76.43%); Dreamer: Valence: 77.44%, Arousal 75.26%, Average: 76.35%)}Spatiotemporal


### 4.4. Evalutation of Recognition Models


*Domain Shift Effects*


EEG signals are intrinsically noisy and vary across subjects, sessions, and equipment leading to a poor signal-to-noise ratio (SNR) and non-unique data distribution for a specific cognitive process. In consideration of the abovementioned, subject-dependent recognition experiments should be able to outperform the subject-independent experiments [28,188]. Building a model for a specific subject reduces the SNR and improves the uniqueness of the data features pattern for specific cognitive processes, e.g., emotion traits. Thus, specific emotions have more distinctive data feature patterns based on the subject’s EEG signal data distribution in subject-dependent experiments. In contrast, in subject-independent experiments, models are required to learn a plethora of data patterns for a specific emotion trait, and the variance and deviation of the data patterns can be significant or contradictory. Consequently, such deviations in the data distribution will result in lower recognition performance. For such changes in data distributions, the defined pattern for an emotion trait will differ between subjects. This change in unique-defined distribution is thus termed a domain shift and is especially pronounced across subjects.

Li et al. [157] identified that this domain shift in the data distribution arose due to variables such as individuals, sex, culture, and genetics. The neural activity has been described to be different across individuals [210] since these individuals modulate their emotions differently from each other [211,212]. However, Lahnakoski et al. argued that brain activity across individuals with the same psychological perspectives is an essential neural process for a shared understanding of the environment [213]. The results from Lahnakoski et al. suggest that brain patterns could be similar for a group of people with a consensus on how happiness is defined. Given the above, reducing these domain shift effects on the data will be feasible, hence improving the recognition model’s performance.

Firstly, calibration-based [214] and alignment-based methods are used to reduce the differences in the distribution and improve the unique definition of the EEG signal’s semantic. Alternatively, transfer learning for domain adaptation methods can address the domain shift issues by mapping EEG features from different domains to a common representation space [215]. Some domain adaptation methods include maximum independence domain adaptation (MIDA) and transfer component analysis (TCA), Geodesic Flow Kernel (GFK), Subspace Alignment, and Information Theoretical learning. With the popularity of deep learning models, transfer learning is performed mainly by pretraining and then fine-tuning.

Subsequently, these deep learning models start to incorporate direct domain adaptation approaches. One such approach is by Luo et al. [216], who proposed the Wasserstein generative adversarial network domain adaptation (WGANDA) for improving recognition performance across subjects. The interest in domain-adversarial networks began following the findings from Ganin et al. [217], and this approach is widely considered a domain adaptation technique. By adopting this DAN domain adaptation technique into the recognition model, Li et al. [198] proposed the domain adaptation neural network (DANN), which consists of three components, the feature extractor, label classification, and the domain discriminator—which inherits from the DAN design. Domain adaptation by incorporating DAN into deep learning models has gained traction over the years. For instance, the few-label adversarial domain adaptation (FLADA) designed for target domains with limited labeled data [207] and the task-specific domain adversarial neural network (T-DANN) consider class boundaries and domain shift simultaneously [206].

In addition to DAN approaches, meta-learning techniques such as MUPS-EEG [203] and MAML [204] have accelerated the transfer process and improved adaptation to new subjects. Also, the Standardization-Refinement Domain Adaptation (SRDA) method utilizes adaptive batch normalization and a loss function based on the variation in information to increase the similarity of marginal distributions between source and target domains [218]. Further analysis of EEG transfer learning techniques can be referred from Wan et al. [219], which provides the analysis for addressing the domain shift problem between subjects.




*PSD and DE Feature Selection*


This section explores how feature selection plays a significant role in determining emotion recognition performance. Soleymani et al. found that the most distinguishable feature of arousal in emotion recognition is the power spectral density (PSD) extracted from the low-frequency Alpha rhythm in occipital EEGs. For valence, key features are primarily from the Beta and Gamma rhythms in temporal lobe EEGs [220]. For analysis of stable EEG activity patterns for different subjects and time periods, Zheng and Lu [151] discovered that the Beta and Gamma rhythms in both sides of the temporal lobes exhibited more substantial activation under positive emotions compared to negative. From Figure 4, there is a trend of increasing performance towards the higher frequency band. Ultimately, it is within the expectation that with more information, the performance of the category of all bands will start to outperform the individual frequency band. Thus, from the feature processing of PSD or DE approaches, coupled with proper frequency band selection, it becomes possible to improve the classification performance.

In addition to band selection, the studies [151,220] identified that occipital and temporal lobes are effective for emotion recognition due to substantial activation. In general, locations associated with the amygdala, frontal, and prefrontal cortex are correlated with emotions, and these channels are identified as temporal: [T7, T8, TP7, TP8], frontal: [F7, F5, F3, F1, FZ, F2, F4, F6, F8], and prefrontal: [AF3, Fp1, FP2, FPZ, AF4].

## 5. Discussion

At this juncture in the review, we have shared a basic overview of the most recent knowledge and insights gleaned from both the neuroscience and affective computing spheres of work.

This review is by no means pitting the work of one field against the other, i.e., brain against computer, nor is this review clamoring for computers to operate like the brain. Hence, even if the brain leverages on oscillating bands to control for the transmission of inputs, it does not imply that computer algorithms should be designed in the same way. Instead, the main takeaways we would like readers to have from all the content prior to this are the following:To understand the “form” behind the “function”: given all the limitations in its basic components (i.e., neurons packed in limited physical space, the need for connectivity and computation efficiency, the presence of conduction delays over axonal branches, etc.), how the brain is eventually structured to overcome those limitations while maintaining an efficient communication mechanism.To understand that brain states are dynamic: functional membership of a neuron can change from one network pool to another under the influence of perturbation from an external stimulus and/or intrinsic population-wide neural activity that is, in turn, governed by different bands of oscillating rhythms.To know the advantages and limitations of emotion-related physiological markers: ERPs, alpha, beta, theta, delta, and gamma frequency bands and the information that can be interpreted or decoded from them.To avail oneself of the latest and widely adopted computational methods, algorithms, and models by the affective computing community for emotion recognition tasks.

A common misconception is that the difference between pre- and post-stimulus EEG signals reflects the properties of the presented stimulus in the brain. In reality, taking a certain statistical difference between baseline and post-stimulus signal reflects how the brain interprets the presented stimulus given its internal state at that point in time. This internal state is highly dependent on current global brain dynamics and regional network microstates that are in turn dependent on modulatory effects from the attentional network and learned priors established from memory. Hence, it would potentially be helpful to view emotion recognition as a framework of a tripartite system: attention–memory–emotion.




*Attention*


Emotions belong to a special class of stimuli that is evolutionarily salient in humans. The fact that the fusiform face area exists specifically for face detection, along with a whole network of differential processes dedicated towards attending, recognizing, and perceiving emotions is ample evidence of its importance. As was mentioned in the subsection on evoked oscillations, alpha, beta, theta, and gamma activity have been shown to play distinct roles in the domain of attention. Alpha functions as the inhibitory gating mechanism that relocates resources away from distracting stimuli; beta functions to resolve conflict in incongruent events while enhancing attention resources towards target stimuli; gamma activity is likely synchronized with beta in an attended task and has been ascribed the function of “feature-binding”; while theta, implicated in multiple functional networks including working memory, provides the gateway for emotion recognition via accessing one’s priors as well as being the acting medium upon which faster oscillations like alpha, beta, and gamma synchronize for sensory encoding.

Together as a small-world dynamic system on its own, the attentional network composed of these four oscillatory bands activates different subsets of neural populations across different brain regions to ultimately result in the desired, successful behavioral output we can observe—guiding us to attend to the desired emotion-expressing target.




*Memory*


Memories can be said to be akin to one’s identity; for without them, we would be without individuality. It is our different experiences-turned-memories that make each of us unique personalities. Inevitably, with subjective experiences comes subjective judgments, and that applies to emotion recognition too. Hence, it should come as no surprise that one’s memory can modulate our emotions and our perception of them.

Behaviorally, the fact that human subjects pass different judgments with different reaction times in emotion change detection tasks [100,101] across varying modalities bears evidence that emotion recognition, which pivots upon subjective memories, is subjective. On top of which, this subjectivity is also affected by the ever-changing internal state of the brain. This is because, structurally, each neuron is connected to another via rules that govern their firing activity. Depending on the properties of the presented stimulus and the shared history of firing patterns with the network they are wired into, large functional networks eventually activate and fire together in fixed rhythmic patterns for prolonged periods of time. This is evidence of the brain’s plasticity at work and that networks do maintain firing patterns based on the ‘memories’ of them.

Such plastic connectivity structures scale up to larger-scale anatomical connectivity that gives rise to certain behavioral responses we see everyday. One extreme example is seen in patients with post-traumatic stress disorder (PTSD). Studies have shown that certain anatomical connections with the amygdala are altered in PTSD patients that are not seen in controls. In general, connectivity between the amygdala and the dorsal anterior cingulate cortex (dACC), a region that is implicated in fear expression [221], as well as that with the hippocampus, a region for memory encoding and retrieval in general [222], are strengthened in PTSD individuals. However, there seems to be a controversy between findings of a weakened [223,224] or strengthened [225,226,227] anatomical connection between the amygdala and ventromedial prefrontal cortex (vmPFC), a center implicated in the regulation of negative-valence emotions. There are several reported findings that can lend moderate support via brain imaging to the functional connectivity between the amygdala, hippocampus, and vmPFC [228,229,230]. Nonetheless, insights from animal studies may be able to provide homologous evidence as inference wherein the hippocampus is found to be involved in both the encoding and retrieval of a contextual fear conditioning episode via genetic manipulations and optogenetics [231,232] and that the mPFC plays a crucial role in the maturation, storage, and recall of remote (long-term) fear memory [63]. In an elegant study conducted by Takashi and his team, they found that with the onset of fear conditioning in the rodent model, information on the negative valence foot shock, encoded via the amygdala, and the contextual information of that fearful episode, encoded via the hippocampus, were transmitted in parallel to the mPFC to first establish immature engram cells (principal neurons that store episodic memories for subsequent retrieval). Recent recall of the fearful episode would trigger activity in both the hippocampus and amygdala but not the mPFC. Over time, the mPFC engrams undergo memory consolidation via reinforcements from the hippocampus and amygdala, respectively, to form long-term memories when the engrams mature. Once matured, remote recall of this long-term, consolidated memory will be done via the mPFC while reactivating the amygdala simultaneously [63]. Although the subregion investigated here is mPFC and not vmPFC per se, it can be inferred that within the larger PFC itself, there is very likely a direct or indirect connection between these two neural populations that are implicated in long-term memory storage and recall and negative-valence regulation.

Given that the amygdala, hippocampus, and PFC are functional components in memory encoding and retrieval of emotionally salient episodes and that studies done in these respective subcortical nuclei have implicated the theta frequency [97,98,119,233,234], it would not be inconceivable to extrapolate that the theta frequency is the medium upon which they communicate. The fact that higher-order cognitive regions such as the PFC have been demonstrated to be involved in multiple processes such as memory and emotion recognition, it emphasizes the importance of slow oscillating bands like theta frequency in the brain for efficient information transmission across various neural network entities.

More importantly, all of these—interplay of various oscillating bands (alpha, beta, theta, etc.) and anatomical brain regions—that are involved in both emotional processing and other cognitive domains such as attention and memory prove that emotional processing (and therefore the study of emotion recognition) is not that straightforward at all. In fact, not one domain can be completely divorced from the other two since they share common brain regions/neural populations and mechanisms of communication. The result of which is manifested in our unique, yet subjective perspective of the world—and in the context of this paper, the way we perceive emotions. How extensive a smile should look like before it is deemed as a smile of happiness; how deep should your brows furrow and crease to signal anger: all of which are heavily anchored by our subjective experiences, stored, and retrieved as memory, and dependent on their saliency or our volition to attend to them in good time.




*Corroborating across Two Fields: Neuroscience and Affective Computing*


In recent years of research (as detailed in Section 4), there has been increasing interest in understanding the mechanisms of the brain and adapting some of the way it does certain computations so that artificial networks mimic a similar operation. Crucially, researchers have recognized that the brain is a dynamic system and that the interpretability of single-subject data is subjected to his/her intrinsic “brain noise”, which stems from a combination of neuron-level (resulting in the non-stationarity problem in EEG data) and functional domain-level interferences (resulting in the domain shift in data distribution). Hence, the analysis of a single cognitive domain, e.g., emotion recognition, will inevitably imply the cross-analysis of other functionally associative domains, e.g., attention and memory. In addition, there is also a need to overcome the brain’s inherent noise in the data in order to make accurate predictions and classifications.

Efforts in this respect include measures such as the online adaptive learning strategy, which attempts to solve the brain’s inverse problem by utilizing its “recent history” of activity as a training set to predict and classify for the subsequent data segment. As mentioned, since EEG readouts of a stimulus from the brain are not a reflection of the stimulus per se but an interpretation of the brain’s internal state having been stimulated by the input, it is imperative to leverage the brain’s recent history in order to extrapolate and predict its present state. The online adaptive learning strategy does exactly that; hence, it is a reliable approach for rectifying the non-stationarity problem to better understand the data. Another measure adopted by researchers to detangle convoluted from multiple cognitive network domains in the brain is the DAN domain adaption technique. However, such algorithms operate on the premise of the availability of large data samples to account for intra-subject variability when it comes to extracting accurate features for classification. With insufficient data samples for even a single subject to be collected over multiple sessions and days, intra-subject noise arising from the shifting brain states during EEG recording will impact subsequent data analysis adversely. Arguably, the publicly available datasets are large. However, the majority of them are simply subject-extensive, and this encourages the field to take the common approach of averaging or cherry-picking “acceptable samples” across subjects or a subset of those in order to make prominent features stand out before inputting them for feature extraction.

On an encouraging note, feature selection from specific brain oscillations is increasingly adopted from experimental findings from neuroscience. For instance, low-frequency alpha spectral power from the occipital region was utilized as a hallmark feature for determining arousal in emotion recognition, while a combination of beta and gamma activity acquired from the temporal lobe area was used to analyze for emotional valence. While not entirely accurate, as alpha activity has been shown to also be involved in valence encoding [105,117] and beta is the band found to be sensitive to arousal values [131], it is still acceptable within limits, which an algorithm can “learn” from the brain. Nonetheless, one recent study did provide some evidence that a decrease in alpha power correlates to high-arousal intensity, although there was also a corresponding decrease in beta power as well. As mentioned earlier in this section, it is not in our interest to push computational systems to completely mimic the brain as it is. Hence, given the premises that (1) alpha and beta are both implicated in the attentional network for stimulus saliency, (2) alpha’s main role serves to diminish effects from distractors while enhancing that for targets hence indirectly implying that arousing components potentially garner attention allocation in the brain, and (3) assuming the objective is to simply classify emotions based on arousal intensity regardless of valence, then alpha power changes could probably serve as the indicator for it. Importantly, it is only justified to leverage changes in alpha power to differentiate arousal intensity for emotion recognition when valence is not taken into account. Should the objective of the classification task include valence values, e.g., high arousal, negative valence, then alpha power alone will not suffice as the sole feature. Other features such as changes in beta and gamma power will be able to serve such classification tasks more efficiently.

Increasingly, the deep learning approach has been gaining traction in affective computing, and it is probably the most “brain-like” of the available approaches—models learn directly from EEG data without the need for handcrafted signals. Moreover, it was shown that the best classification results are obtained from using information from all five frequency bands in a short time window (not exceeding 5 s). Since the brain constantly has ongoing activity across all five frequency bands that hold rich information on its internal state, the model recognizes value in not discriminating or cherry-picking specific frequency bands for analysis in order to obtain the maximum breadth of information in a particular instant. In addition, a short sampling time window helps to preserve information in the gamma frequency band. Since gamma represents transient, local computations that could facilitate “feature-binding” across different brain regions and functional domains [98,99,139] while being modulated by slower oscillating frequencies, gamma likely contains important information that pertains to early sensory processing for encoding of stimuli, especially when accounted for in a spatial-specific manner on the scalp. Although it is yet to be seen if a standalone gamma activity analysis will be able to provide robust and reliable information for emotion recognition, analyzing its interaction and cross-coupling effects with other frequency bands will definitely yield functionally meaningful interpretations.

## 6. Conclusions

Emotional intelligence is a multifaceted domain of study, and understanding it in its entirety will require investigations on the generation, detection, and recognition of emotion (Figure 5). With cross references and a combination of various applied theories, data collection modalities, and analyses, we are seeing a burgeoning of novel findings from the fields of neuroscience, engineering, and computer science—most of which are leveraging the brain’s oscillatory features and some of their hallmark biomarkers, e.g., ERPs as probes for decoding emotion.

Despite the excellent work and progress being made over the past decade, findings have been correlational at best, and that models that have been generated are mostly to “mimic” some, if not all, of the biological features mined from varied definitions and classes of emotions. Admittedly, the quest for causality in this aspect is a hard problem because it will involve identifying the brain’s oscillatory sources, ablating those sources, and accounting for the recalibration of time constants from remnant neurons and the associated excitatory and inhibitory interactions (as these time constants and neuronal connections serve as inherent oscillatory sources) before one can begin to investigate how a certain oscillatory source impacts both local and global neural circuitries in the brain. Consequently, such a task is deemed impossible since tampering with oscillations will likely cause the biosystem to malfunction altogether, thereby rendering further investigations with the altered biosystem impossible. A remote example will be that of a stroke patient—wherein a lesion in the brain alters the inherent oscillatory rhythm of the brain such that the remaining intact neural constituents undergo cortical reorganization to attain a “new equilibrium” that would allow the system to preserve as much of its functionalities as possible.

One suggestion to approach understanding brain oscillations is to take a different stance to how oscillations are being perceived. Unlike the relatively “tangible” brain components such as cortical subregions or specific clusters of neurons identified with a certain function, oscillations should be considered as an “intangible” property that simply indicates how the brain is keeping track of time. The rhythm to which the brain uses to control how one region communicates to another can be a superposition of many different frequencies (i.e., a combination of different paces) all at once. Hence, in terms of biology, there is no one brain region or body of cells that can be labeled as “top of the feed-forward hierarchy” since the back-and-forth communication kept in check via oscillations ensures that there is no actual beginning or end in the transmission chain(s).

Therefore, it is potentially of greater value to investigate brain oscillations and the resultant effects they generate on the behavioral level (i.e., emotions) by manipulating the temporal consequences that inputs, outputs, and their intermediaries may have on the local and global scale from a certain neural circuitry, e.g., for the purpose of building effective brain–computer interfaces (BCI), human–computer interfaces (HCI), and socially behaving robots—something that researchers from various fields can work towards in the near future. From there, the next breakthrough would likely manifest in the context of understanding and modeling emotion regulation, which is prevalent in a multitude of psychological disorders that may or may not stem from neurological dysfunction.

## Figures and Tables

**Figure 1 brainsci-14-00364-f001:**
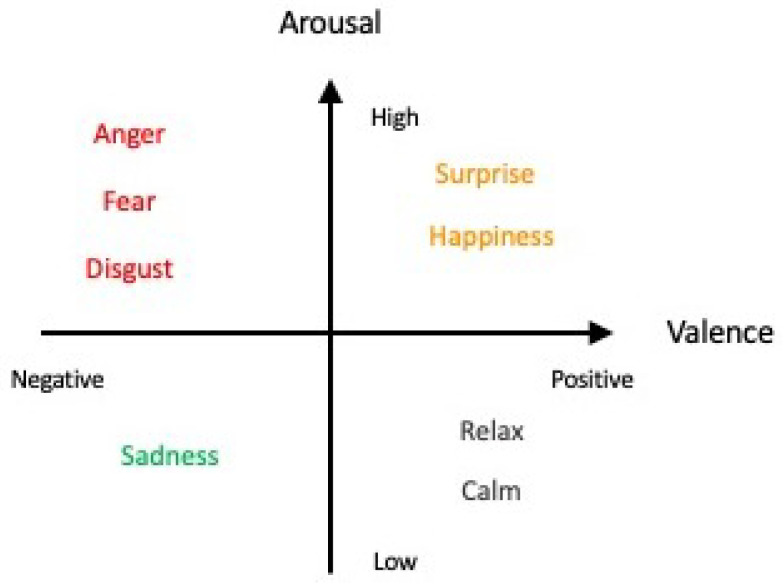
Affective model with the six fundamental classes of emotion in colored font (adapted from Russell, 1980 [7]).

**Figure 2 brainsci-14-00364-f002:**
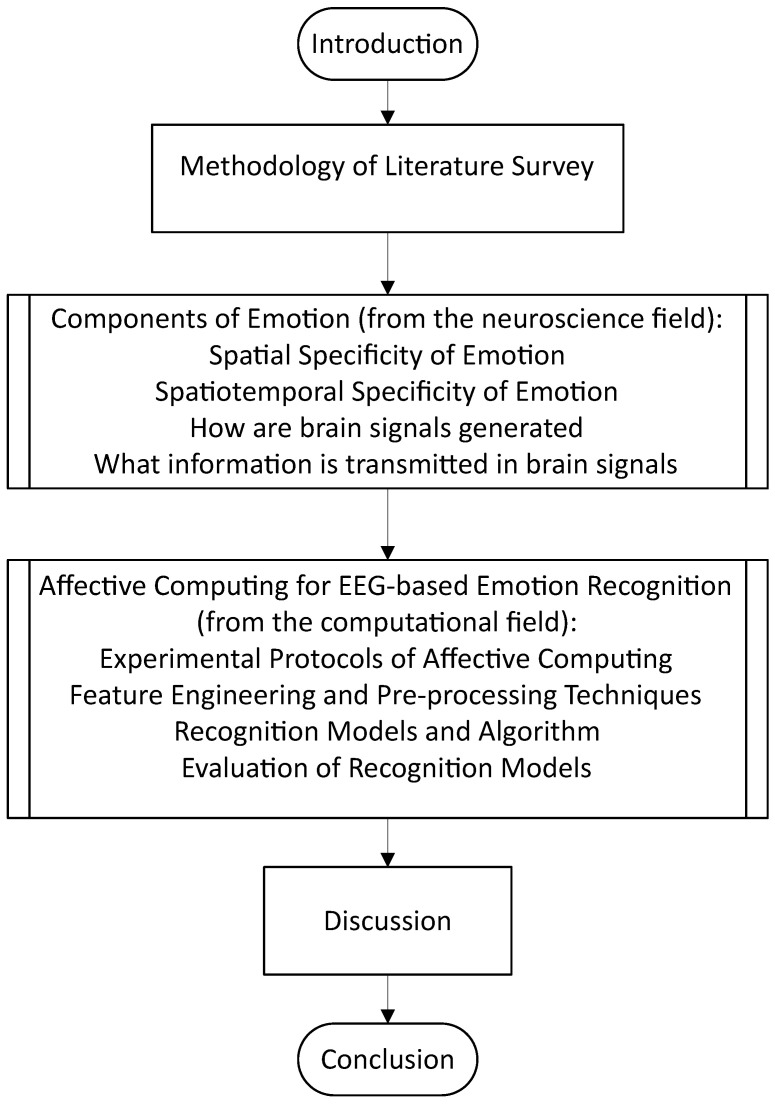
Overview of the review article.

**Figure 3 brainsci-14-00364-f003:**
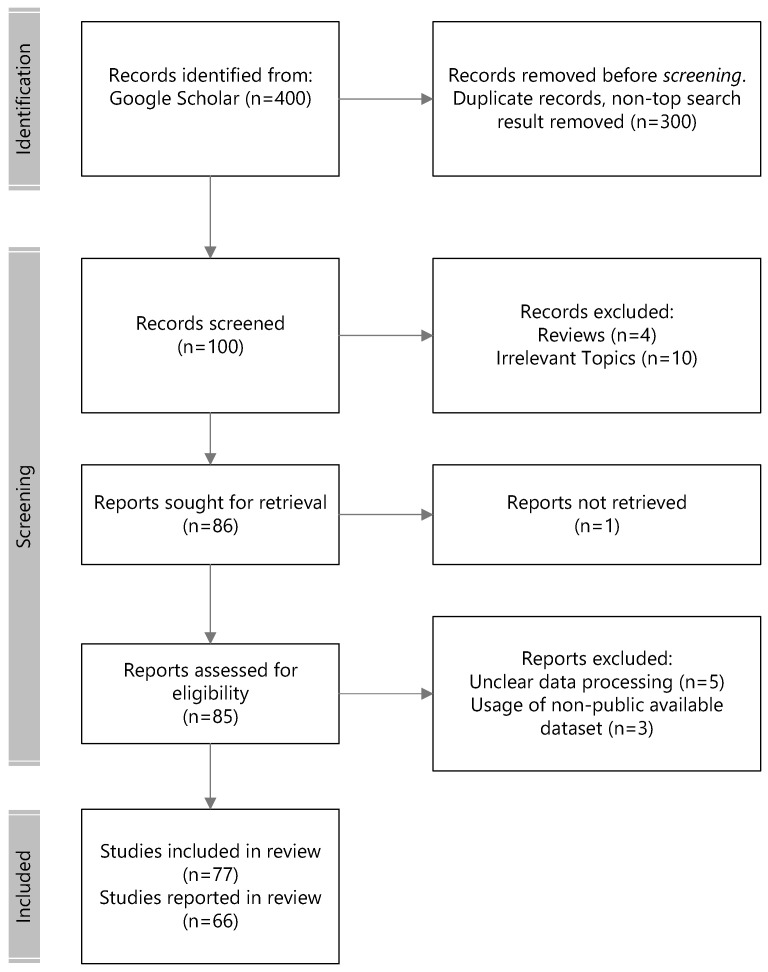
Flowchart of the literature selection.

**Figure 4 brainsci-14-00364-f004:**
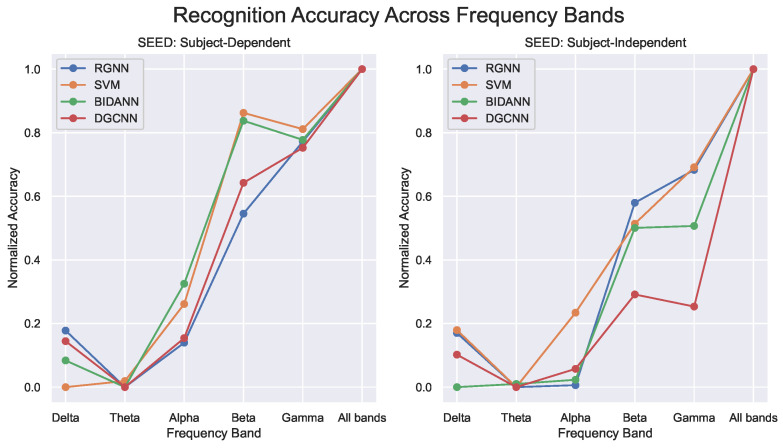
Normalized classification performance across frequency bands and entire bands. Accuracy scores are derived from [188,189,201].

**Figure 5 brainsci-14-00364-f005:**
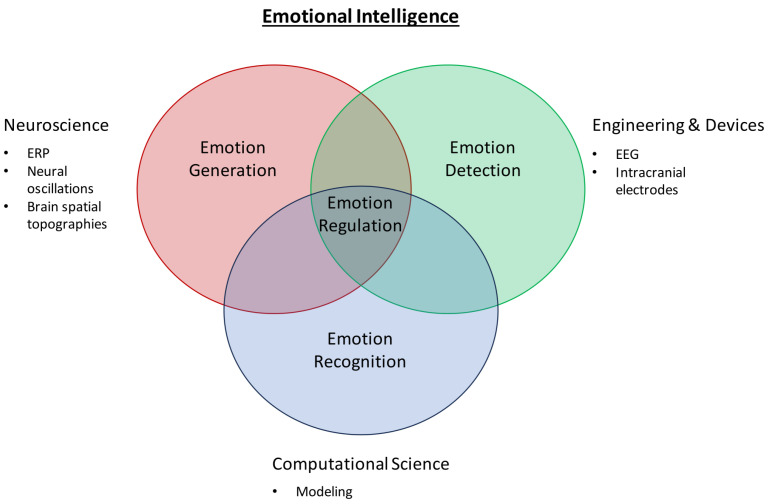
Venn diagram illustrating the interdisciplinary fields for emotion recognition: neuroscience, engineering and devices, and computational science.

**Table 1 brainsci-14-00364-t001:** Emotions and their associated brain regions.

Emotion	Study	Activated Brain Region(s)	Stimulus Modality	Imaging Modality
Happiness	Habel et al., 2005 [30]	LEFT dorsolateral prefrontal cortex, LEFT anterior cingulate gyrus, RIGHT cingulate gyrus, LEFT posterior cingulate gyrus, LEFT middle temporal gyrus, LEFT parietal cortex, LEFT angular gyrus, RIGHT paracentral lobule, LEFT parahippocampal gyrus, LEFT amygdala	Visual (static images)	fMRI
	Killgore and Yurgelun- Todd, 2004 [31]	LEFT/RIGHT amygdala, LEFT/RIGHT anterior cingulate	Visual (static images)	fMRI
	Esslen et al., 2004 [32]	LEFT/RIGHT frontal cortex, LEFT/RIGHT ventromedial frontal, LEFT/RIGHT temporal cortex, RIGHT parietal area, anterior cingulate cortex, RIGHT frontal areas	Visual (static images)	EEG (LORETA)
	Gorno-Tempini et al., 2001 [33]	RIGHT precentral sulcus, RIGHT middle and inferior frontal gyri, RIGHT posterior fusiform gyrus, RIGHT anterior insula, occipito-temporal cortex, prefrontal cortex, amygdala, basal ganglia, LEFT/RIGHT orbitofrontal cortex	Visual (static images)	fMRI
	Kesler-West et al., 2001 [34]	LEFT/RIGHT fusiform gyri, LEFT/RIGHT amygdala, LEFT/RIGHT entorhinal cortices, RIGHT superior temporal sulcus, RIGHT inferior occipital gyrus, LEFT/RIGHT inferior frontal gyri, RIGHT angular gyrus, LEFT/RIGHT lingual gyri, medial frontal/cingulate sulcus	Visual (static images)	fMRI
	van de Riet et al., 2009 [35]	calcarine sulcus, anterior cingulate gyrus, cerebellum, RIGHT fusiform gyrus, parietal lobe, primary motor cortex, premotor cortex, primary somatosensory cortex, olfactory sulcus, insula	Visual (static images of faces and bodies)	fMRI
	Fitzgerald et al., 2006 [36]	LEFT amygdala, LEFT lingual gyrus	Visual (static images of faces)	fMRI
Sadness	Habel et al., 2005 [30]	LEFT dorsolateral prefrontal cortex, LEFT orbitofrontal cortex, LEFT superior frontal gyrus, LEFT middle temporal gyrus, LEFT superior temporal gyrus, LEFT precuneus, LEFT parahippocampal gyrus, LEFT amygdala-hippocampal area, LEFT putamen, RIGHT fasciculus occipito-frontalis, LEFT insula	Visual (static images of faces)	fMRI
	Killgore and Yurgelun-Todd, 2004 [31]	LEFT/RIGHT amygdala, LEFT/RIGHT anterior cingulate	Visual (static images of faces)	fMRI
	Esslen et al., 2004 [32]	LEFT postcentral area, RIGHT prefrontal cortex, LEFT/RIGHT ventromedial frontal, LEFT/RIGHT orbitofrontal, RIGHT prefrontal cortex, LEFT/RIGHT temporal cortex, RIGHT occipital cortex, LEFT superior frontal cortex, RIGHT frontal cortex, LEFT/RIGHT posterior cingulate cortex, LEFT/RIGHT occipital lobe	Visual (static images of faces)	EEG (LORETA)
	Kesler-West et al., 2001 [34]	LEFT/RIGHT fusiform gyri, LEFT/RIGHT amygdala, LEFT/RIGHT entorhinal cortices, RIGHT superior temporal sulcus, RIGHT inferior occipital gyrus, LEFT/RIGHT inferior frontal gyri, RIGHT angular gyrus, LEFT/RIGHT lingual gyri	Visual (static images of faces)	fMRI
	Fitzgerald et al., 2006 [36]	LEFT amygdala, RIGHT middle temporal gyrus, LEFT middle frontal gyrus, RIGHT inferior frontal gyrus	Visual (static images of faces)	fMRI
Fear	Esslen et al., 2004 [32]	LEFT temporal area, RIGHT frontal pole, RIGHT temporal cortex	Visual (static images of faces)	EEG (LORETA)
	Kesler-West et al., 2001 [34]	LEFT/RIGHT fusiform gyri, LEFT/RIGHT amygdala, LEFT/RIGHT entorhinal cortices, RIGHT superior temporal sulcus, RIGHT inferior occipital gyrus, LEFT/RIGHT inferior frontal gyri, RIGHT angular gyrus, LEFT/RIGHT lingual gyri	Visual (static images of faces)	fMRI
	van de Riet et al., 2009 [35]	calcarine sulcus, anterior cingulate gyrus, cerebellum, LEFT/RIGHT amygdala, LEFT/RIGHT fusiform gyri, RIGHT inferior occipital gyrus, RIGHT superior temporal sulcus, superior colliculus	Visual (static images of faces and bodies)	fMRI
	Fitzgerald et al., 2006 [36]	LEFT amygdala, LEFT inferior frontal gyrus	Visual (static images of faces)	fMRI
	Lange et al., 2003 [37]	LEFT amygdala, LEFT hippocampus, LEFT putamen, LEFT ventrofrontal gyrus, LEFT lingual gyrus, LEFT/RIGHT cerebellum	Visual (static images of faces)	fMRI
	Liddell et al., 2005 [38]	LEFT/RIGHT Amygdala, LEFT pulvinar, LEFT superior colliculus, LEFT locus coeruleus, LEFT/RIGHT anterior cingulate, LEFT inferior frontal gyrus, RIGHT middle frontal gyrus, LEFT cingulate gyrus, LEFT medial frontal, LEFT superior/middle frontal gyri, LEFT caudate, LEFT insula, LEFT postcentral gyrus, LEFT superior temporal gyrus, LEFT middle temporal gyrus, LEFT inferior temporal gyrus	Visual (static images of faces)	fMRI
	Phillips et al., 2004 [39]	LEFT/RIGHT posterior cingulate gyrus, LEFT putamen, RIGHT inferior parietal lobule, RIGHT amygdala/hippocampus, RIGHT anterior cingulate gyrus, RIGHT lingual gyrus, RIGHT medial frontal gyrus, RIGHT precuneus, RIGHT cerebellum, LEFT inferior frontal gyrus, LEFT/RIGHT superior temporal gyrus, RIGHT middle temporal gyrus	Visual (static images of faces)	fMRI
	Thielscher & Pessoa, 2007 [40]	LEFT lingual gyrus, RIGHT precuneus, LEFT fusiform gyrus, RIGHT entorhinal cortex, LEFT/RIGHT anterior insula/inferior frontal gyrus, RIGHT inferior frontal sulcus, RIGHT orbital gyrus, RIGHT thalamus, LEFT caudate	Visual (static images of faces)	fMRI
	Williams et al., 2005 [41]	LEFT amygdala, LEFT hippocampus, LEFT/RIGHT medial prefrontal cortex, RIGHT anterior cingulate, RIGHT lateral prefrontal cortex, RIGHT occipital gyrus, RIGHT fusiform gyrus, LEFT superior temporal gyrus, LEFT/RIGHT postcentral gyrus, LEFT putamen, LEFT thalamus	Visual (static images of faces)	fMRI
Anger	Esslen et al., 2004 [32]	RIGHT frontal lobe, MEDIAL superior frontal gyrus, RIGHT prefrontal cortex	Visual (static images of faces)	EEG (LORETA)
	Kesler-West et al., 2001 [34]	LEFT/RIGHT fusiform gyri, LEFT/RIGHT amygdala, LEFT/RIGHT entorhinal cortices, RIGHT superior temporal sulcus, RIGHT inferior occipital gyrus, LEFT/RIGHT inferior frontal gyri, RIGHT angular gyrus, LEFT/RIGHT lingual gyri, LEFT precentral gyrus, medial superior frontal gyrus, RIGHT lateral occipital gyrus	Visual (static images of faces)	fMRI
	Fitzgerald et al., 2006 [36]	LEFT amygdala, LEFT inferior temporal gyrus, LEFT fusiform gyrus, LEFT middle frontal gyrus	Visual (static images of faces)	fMRI
	Williams et al., 2005 [41]	RIGHT amygdala, LEFT/RIGHT medial prefrontal cortex, RIGHT anterior cingulate, LEFT lateral prefrontal cortex, LEFT occipital gyrus, LEFT fusiform gyrus, RIGHT superior temporal gyrus, RIGHT putamen, RIGHT thalamus	Visual (static images of faces)	fMRI
Disgust	Esslen et al., 2004 [32]	RIGHT frontal cortex, LEFT/RIGHT frontal lobes, anterior cingulate cortex, premotor cortex	Visual (static images of faces)	EEG (LORETA)
	Gorno-Tempini et al., 2001 [33]	RIGHT precentral sulcus, RIGHT middle and inferior frontal gyri, RIGHT posterior fusiform gyrus, RIGHT anterior insula, occipito-temporal cortex, prefrontal cortex, LEFT amygdala, basal ganglia, RIGHT striatum, RIGHT head of caudate nucleus, RIGHT thalamus	Visual (static images of faces)	fMRI
	Fitzgerald et al., 2006 [36]	LEFT amygdala, LEFT inferior occipital gyrus, LEFT middle occipital gyrus, LEFT medial frontal gyrus, LEFT lingual gyrus, LEFT inferior frontal gyrus, LEFT medial frontal gyrus, LEFT inferior frontal gyrus	Visual (static images of faces)	fMRI
	Phillips et al., 2004 [39]	LEFT middle temporal gyrus, LEFT/RIGHT posterior cingulate gyrus, RIGHT precuneus, LEFT/RIGHT cerebellum, LEFT postcentral gyrus, LEFT inferior parietal lobule, RIGHT insula, RIGHT anterior cingulate gyrus, RIGHT lingual gyrus, RIGHT superior temporal gyrus, LEFT middle occipital gyrus	Visual (static images of faces)	fMRI
	Thielscher & Pessoa, 2007 [40]	RIGHT middle occipital gyrus, LEFT intraparietal sulcus, RIGHT middle temporal gyrus	Visual (static images of faces)	fMRI
	Williams et al., 2005 [41]	LEFT/RIGHT insula, LEFT/RIGHT amygdala-hippocampal complex, RIGHT hippocampal gyrus, RIGHT medial prefrontal cortex, LEFT/RIGHT anterior cingulate, LEFT/RIGHT lateral prefrontal cortex, RIGHT occipital gyrus, LEFT/RIGHT superior temporal gyrus	Visual (static images of faces)	fMRI

**Table 4 brainsci-14-00364-t004:** Benchmarked EEG datasets.

Year	Dataset	Stimuli	Emotion Model	Modalities Type	EEG Stats (Channels, Sampling Frequency, etc.)	Participants	EEG Device
2012	DEAP [149]	40 Video clips— Music Video	Valence, Arousal, Dominance, Liking, Familiarity	EEG, EOG, EMG, Galvanic Skin Response, Respiration, Temperature, Plethysmograph.	32 EEG channels, 12 peripheral channels, Rae Signals—512 Hz, Preprocessed data: 128 Hz)	32 (16 male, 16 female), Race: Mixed	Biosemi ActiveTwo
2012	MAHNOB HCI [150]	20 Video Clips	Dimensional: Arousal, Valence, Control, Predictability. Emotion	EEG, EXG, GSR, Respiration, Temperature, Eye Gaze	32 Channels	27 (11male, 16 female), Race: Mixed	Biosemi ActiveTwo
2015	SEED [151]	15 Video Clips	Positive, Negative, Neutral	EEG, EMG, EOG	62 Channels	15 (7 male, 8 female), Race: Chinese	ESI NeuroScan
2018	Dreamer [152]	18 Video Clips	Arousal, Valence, Dominance	EEG, ECG	14 Channels	25 (14 male, 11 female), Race: Mixed	Emotiv EPOC
2019	SEED-IV [153]	72 Video Clips	Happy, Sad, Fear and Neutral	EEG, Physiological Signals	62 Channels	15 (7 male, 8 female), Race: Chinese	ESI Neuroscan
2019	MPED [154]	28 Video Clips	Joy, Funny, Anger, Fear, Disgust, sad, neutral	EEG, ECG, GSR, Respiration	62 Channels	23 (10 male, 13 female), Race: Chinese	ESI NeuroScan
2021	AMIGOS [155]	16 Videos	Big-Five Personality traits, Emotions; Valence Arousal, Dominance, Liking, Familiarity	EEG, ECG, GSR, Audio Visual, Depth	14 Channels, 128 Hz	40 (27 male, 13 female) for short videos; 37 for Long Videos, Race: Mixed	Emotiv EPOC NEuroheadset
2022	SEED-V [156]	15 Videos	Happy, Sad, Fear, Neutral, Disgust	EEG, Physiological Signals	62 Channels	20 (10 male, 10 female), Race: Chinese	ESI NeuroScan

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
