# Peer review of "Review of EEG Affective Recognition with a Neuroscience Perspective"

_brainsci, 2024, doi:10.3390/brainsci14040364_

Round 1

Reviewer 1 Report

Comments and Suggestions for Authors

Emotions are a range of transient, subconscious, and even elusive ways that the human inherent system manifests itself. They have a significant impact on how we perceive ourselves, our environment, and how we engage with it in daily life. Many studies have been conducted to date in the fields of affective computing and neuroscience, using neural network models and experimental data, respectively, to clarify the brain circuitry and neural correlates for emotion recognition. Neuroscientific findings and viewpoints are frequently strongly linked to recent developments in emotional computing neural network models. In particular, there is a rising interest in using models based on the neurological foundations of the generation, processing, and subsequent collecting of EEG data for EEG-based emotion identification. In this regard, the review concentrates on offering perspectives and neuroscientific evidence to address how emotions might arise as a result of neural activity at the level of subcortical structures within the brain's emotion circuitry, as well as the connection to existing affective computing models for emotion recognition. We will also talk about whether this kind of biologically inspired modeling may help improve the field of EEG-based emotion identification and other related fields.

- how the authors have made figure three since no data was collected in this work? 

- It seems that the area covered in this article is exhaustive which need a conciseness. 

- How the authors have searched several articles in relation to affectiveness. Such criteria, must be included to the mansucript. 

- The last paragraph of Introduction did not introduce Section 3. I suggest the authors put a figure of organization of this work. Or put a figure to describe the relationship between all the contents including each section. Then readers can have a general organization of this survey.
- The authos must add some studies related to the hybrid techqnieues along with EEG so that better results can be acheived in terms of affective

- Few more applications should be added to the manuscript. 

Reviewer 2 Report

Comments and Suggestions for Authors

This review focuses on understanding neural circuitry and correlates, aiming to advance emotion recognition models. It also assesses the efficacy of biologically-inspired modelling for EEG-based emotion recognition. However, the following suggestions are required to improve the readability of the manuscript.

1.       In the first paragraph of the Introduction, Include the WHO recent statistics emotional related mental health disorders are required.

2.       In the second paragraph of introduction only two methods are considered, Emotions also can be described using PAD model: http://dx.doi.org/10.3389/fpsyg.2022.942198

3.       Also, the review does not provide the comparison of these methods and explain which one is widely is preferred and why?

4.       Mention all physiological and non-physiological signals used for emotion recognition. You could use the following reference A Systematic Review of Sensing and Differentiating Dichotomous Emotional States Using Audio-Visual Stimuli. It also explores EEG studies and their instrumentations for classifying Happy and sad emotional states.

5.       In Introduction, paragraph 3 does not have any references.

6.       What are the advantages of using EEG for emotion recognition is not highlighted over other physiological signals like EDA?

7.       Make the manuscript more technical, It looks like an AI generated text.

8.       Physiology behind generation of EEG is missing. Write a paragraph on this. Also highlight the brain regions involved in various emotional states.

9.       There are also studies on correlation of EEG and EDA signals..Include this concept as a paragraph before starting the previous works in the introduction: https://doi.org/10.1109/BHI.2019.8834567 ; https://doi.org/10.1109/ACII.2013.162

10.   Include a paragraph on How EEG is generated during emotions, electrodes used, Brodman regions involved during emotions…Also EEG caps available and which is widely used.

11.  Include a paragraph on channel selection criteria, and which channels are effective in emotions

12.  Include recent works available in all domains: Time, Frequency, TF

13.  What is recent state-of-the-art of deep learning based Emotion recognition using EEG? Write a separate paragraph.

14.   Why there is  a question mark after reference 20?

15.  References are missing for many statements. Check the manuscript thoroughly and add the referencs

16.  Highlight the need for and importance of this review in a separate paragraph as a last paragraph of introduction.

17.  Include more effective databases: Scopus, IEEE, Pubmed and rewrite the article based on the articles. Also provide the search terms used in each database as a supplementary file. If we paste the search terms in the database, the article numbers has to be matched with the numbers you have mentioned in the manuscript.

18.  Create a proper flow to the manuscript. In components of emotion section, why you have included Table 1. Does it make sense?

19.  The literature included in the manuscript need to be improved. Including the current state-of-the-art works in emotion recognition such as: https://doi.org/10.1109/JSEN.2024.3354553 will improve the readability of the paper. These methods could be best for EEG based emotion recognition, since EEG also a non-linear signal. Include them as a recent advances for future work.

20.  Explore the recent deep learning methods also for emotion recognition, especially methods like VFCDM+CNN for emotion recognition could be potential future work which was explored for epilepsity dection using EEG signals. Use this reference als in the manuscript EEG-Based Seizure Detection Using Variable-Frequency Complex Demodulation and Convolutional Neural Networks.

21.  In feature Engineering: Time domain, The Hjorth parameters are modified and two more parameters are described. Non-Parametric Classifiers Based Emotion Classification Using Electrodermal Activity and Modified Hjorth Features. Please refer this and use in the manuscript.

22.  In TF domain, VFCDM could be used

23.  In deep learning, VFCDM + CNN combination could a potential future work

Comments on the Quality of English Language

NA

Round 2

Reviewer 1 Report

Comments and Suggestions for Authors

the authors have addressed my comments.

Reviewer 2 Report

Comments and Suggestions for Authors

Authors have incorporated all of my suggestions

Comments on the Quality of English Language

NA